# SpecMER: Fast Protein Generation with K-mer Guided Speculative Decoding

**Thomas A. Walton**
Georgia Institute of Technology
twalton42@gatech.edu

**Darin Tsui**
Georgia Institute of Technology
darint@gatech.edu

**Aryan Musharaf**
Georgia Institute of Technology
amusharaf3@gatech.edu

**Amirali Aghazadeh**
Georgia Institute of Technology
amiralia@gatech.edu

## Abstract

Autoregressive models have transformed protein engineering by enabling the generation of novel protein sequences beyond those found in nature. However, their sequential inference introduces significant latency, limiting their utility in high-throughput protein screening. Speculative decoding accelerates generation by employing a lightweight draft model to sample tokens, which a larger target model then verifies and refines. Yet, in protein sequence generation, draft models are typically agnostic to the structural and functional constraints of the target protein, leading to biologically implausible outputs and a shift in the likelihood distribution of generated sequences. We introduce SpecMER (Speculative Decoding via k-mer Guidance), a novel framework that incorporates biological, structural, and functional priors using k-mer motifs extracted from multiple sequence alignments. By scoring candidate sequences in parallel and selecting those most consistent with known biological patterns, SpecMER significantly improves sequence plausibility while retaining the efficiency of speculative decoding. SpecMER achieves 24–32% speedup over standard autoregressive decoding, along with higher acceptance rates and improved sequence likelihoods.

## 1 Introduction

Designing proteins with enhanced or novel biological functions is an important problem with wide-ranging applications in therapeutics, sustainability, and drug discovery [1]. Protein design is challenging due to the astronomically large space of possible protein sequences, of which only a tiny fraction is likely to exhibit the desired functions [2]. Generative models trained on the vast landscape of evolved protein sequences have emerged as powerful tools for navigating this combinatorial sequence space. Among them, autoregressive models have been especially successful in the design of proteins with functional properties comparable to natural proteins [3]. Their autoregressive nature integrates well with textual information [4], ensures training stability [5], enables variable-length sequence generation [5], and effectively captures long-range dependencies that encode complex amino acid interactions [4]. Despite these advantages, autoregressive models struggle with pathological repetition [6–8], early termination [7], out-of-distribution generalization [8], and most consequential, high computational cost during inference, resulting in slow sequence generation. This computational cost is a debilitating limitation for high-throughput protein screening, where thousands of protein sequences need to be generated and screened to create large-scale libraries of functional and structurally stable proteins [9]. For instance, generating 20,000 protein sequences of length 200 amino acids using ProGen2-XL, a 6.4-billion-parameter transformer-based autoregressive model, takes approximately 65 hours using a single NVIDIA A6000 GPU. Such slow sequence generation can

39th Conference on Neural Information Processing Systems (NeurIPS 2025).

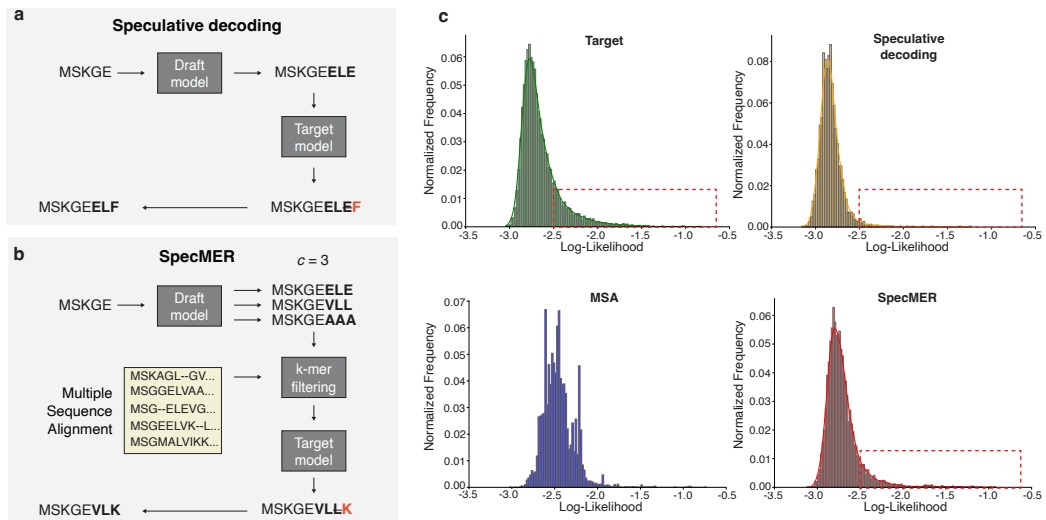

Figure 1: **Overview of speculative decoding and SpecMER. a**, Given a context sequence, speculative decoding uses a draft model to sample tokens. The target model verifies generated tokens, accepting or rejecting them. In this example, two tokens were accepted from the draft model, and the token "$E$" was corrected to "$F$". **b**, SpecMER augments the drafting process, generating multiple sequences which are then filtered by k-mers at near-zero cost. K-mers are formed through a multiple sequence alignment, capturing evolutionary patterns through homologous protein sequences. In this example, "$VLL$" was selected from the proposed candidates, where two tokens were accepted and the token "$L$" was corrected to "$K$". **c**, Speculative decoding undersamples high log-likelihood sequences, missing the tail distribution of the target model. K-mers act as guidance for SpecMER, resulting in a distribution of generated sequences that more adequately samples high-likelihood, high-plausibility protein sequences. Additionally, SpecMER retains nearly the same generation speed as speculative decoding, with only a $3\%$ overhead observed at $c = 3$.

delay high-throughput design workflows by days, not accounting for the additional time required for iterative design processes.

Recently, speculative decoding [10] has proven effective in accelerating autoregressive generation in natural language processing. Speculative decoding utilizes a lightweight *draft* model to propose tokens, which are then verified by a larger, more expressive *target* model. This speculate-then-verify framework of generation has demonstrated wall-time speedup factors exceeding $2\times$ by selecting outputs consistent with the target model distribution [11].

In protein sequence generation, however, target-consistent outputs may not align with the most biologically plausible proteins, particularly those that best satisfy structural or functional constraints. This misalignment is amplified in protein design workflows, where satisfying biochemical constraints is essential. While draft models may capture broad structural patterns, small deviations in local structure can lead to misfolded or nonfunctional proteins. Among target-consistent outputs, some may result in proteins with more plausible biological properties; however, such distinctions are not considered during speculative decoding. To address this, incorporating biologically meaningful local priors offers a principled way to guide generation. Notably, k-mers, or short contiguous amino acid subsequences, encode local structural motifs that correlate with both folding and function, and can be extracted with minimal computational cost.

We hypothesized that incorporating structural cues from k-mers could guide speculative decoding toward more biologically plausible proteins, while retaining or even enhancing generation speed. To test this hypothesis, we develop speculative decoding via k-mer guidance (SpecMER), a novel decoding framework that leverages structural motifs from homologous protein sequences to improve both generation speed and quality. By incorporating structural priors into the decoding process, SpecMER provably selects tokens that are not only consistent with the target model but also biased toward completions likely to yield structured, biologically plausible proteins. This motif-informed

**Algorithm 1** Token-level maximal coupling [11]

**Input:** Distributions $p$, $q$, Draft sample $X \sim p$.

Compute the residual distribution $p_{\text{res}}$ where $\forall x \in \mathcal{V}$, $p_{\text{res}}(x) = \frac{q(x) - \min\{p(x), q(x)\}}{1 - \sum_{x' \in \mathcal{V}} \min\{p(x'), q(x')\}}$.

Sample $\eta \sim U(0, 1)$

**if** $\eta \leq \min\left(1, \frac{q(X)}{p(X)}\right)$ **then**

    **Return** $Y = X$. {Accept the draft token.}

**else**

    **Return** $Y \sim p_{\text{res}}$. {Correct the token by sampling from the residual distribution.}

**end if**

---

guidance increases the likelihood that generated sequences will fold into stable structures. Our contributions are as follows:

- We develop the first speculative decoding framework for protein generation. Using ProGen2-S as the draft model and ProGen2-M as the target model, we demonstrate an average increase in generation speed by $32\%$.

- We develop SpecMER, a novel framework that incorporates k-mer frequencies from natural proteins to guide draft token selection. SpecMER produces sequences more likely to resemble natural proteins while retaining increased generation speed from speculative decoding, enabling fast generation of plausible proteins. Software for SpecMER is available at `https://github.com/amirgroup-codes/SpecMER.git`.

- We establish theoretical bounds on wall-time speedup and characterize how guidance from k-mers improves draft token selection, offering practical guidance for estimating speedups under varying system configurations and sequence constraints.

## 2 Preliminary

**Notation.** Let draft model $P$ and target model $Q$ be autoregressive models with probability density functions $p$ and $q$. A protein sequence $x \sim P$ is denoted by sequence $x(i), x(i+1), \ldots, x(j)$, where $x(i)$ is the $i^{th}$ token sampled from $p$. Tokens are defined by the shared vocabulary $\mathcal{V}$ of $P$ and $Q$, consisting of amino acids and special tokens.

### 2.1 Speculative Decoding

Speculative decoding has recently emerged as a highly effective technique for accelerating autoregressive generation. Speculative decoding employs a smaller draft model to propose candidate tokens, which are then refined by a larger, more expressive target model [10, 11]. This approach generates tokens consistent with the target model distribution while significantly reducing decoding latency. Various extensions to speculative decoding have explored architectural modifications [12], tree-based verification [13], distillation of the draft model [14], and optimized scheduling or reference-free inference [15].

Given a context sequence $x(t)$ and a draft model $P$, speculative decoding efficiently samples $L$ tokens, $\tilde{x}(t+1), \ldots, \tilde{x}(t+L)$. Speculative decoding can be broken down into three steps: 1) Draft construction, 2) Conditional probability computation, and 3) Draft selection (Figure 1a).

1. **Draft construction.** The draft model first samples $L$ tokens, $\tilde{x}(t+1), \ldots, \tilde{x}(t+L)$. For every token $i < L$, we use the draft model to compute the conditional probability of the next token $y$, $P(y \mid x(t), \tilde{x}(t+1 : t+i))$, where $y \in \mathcal{V}$.

2. **Conditional probability computation.** For every token $i < L$, we use the target model $Q$ to compute the conditional probability of the next token $Q(y \mid x(t), \tilde{x}(t+1 : t+i))$.

3. **Draft selection.** We validate the first $L'$ out of $L$ tokens based on the conditional probabilities of the draft and target models. We set $x(t+i) = \tilde{x}(t+i)$ for $i \leq L'$. For the rejected token $\tilde{x}(t+L'+1)$, sample from a residual distribution and correct it.

After steps 1 through 3, we use $x(t + L' + 1)$ as the next context and continue the speculative decoding process iteratively. Algorithm 1 [11] details the acceptance and rejection process for each

token generated by the draft model. Rejected tokens are corrected by sampling from the residual distribution $p_{res}(x)$.

The correctness guarantees of speculative decoding are formally defined for ancestral sampling, where outputs are drawn directly from the full target distribution. In protein language models, however, ancestral sampling is rarely used in practice, as it often produces implausible or repetitive sequences [8]. Instead, decoding with nucleus sampling (top-$p$) is common to balance plausibility and diversity, and is utilized with both ProGen and ProGen2 [4, 8, 16, 17]. When paired with speculative decoding, nucleus sampling truncates the low-probability tail of the target distribution, as observed in Figure 1. Since this tail accounts for a small fraction of probability mass ($p = 0.95$ in this study), outputs remain closely aligned with the target model, with differences confined to rare tail events.

Speculative decoding has demonstrated impressive speedups in natural language generation, where structural constraints are minimal. However, existing methods are agnostic to domains like protein design, where sequences must satisfy intricate biochemical and structural requirements. In this setting, small changes can disrupt folding, stability, or function, and distributional drift between the draft and target models can accumulate over time, degrading generation quality. To the best of our knowledge, no prior work in speculative decoding incorporates sequence-level structural information to guide draft generation, limiting its applicability in this domain. This work addresses that gap by introducing a structure-aware drafting strategy that leverages local sequence motifs to guide speculative decoding toward more plausible protein sequences.

**Which tokens are optimal?** Assume that the acceptance ratio of token $x(t + i)$ sampled from $p$ is given by $\beta(t + i)$. If we assume each $\beta(t + i)$ is i.i.d., then $\mathbb{E}[\beta] = \alpha$, where $\alpha$ indicates the overall acceptance ratio for a given configuration of speculative decoding [10]. It follows from [11] that $\alpha$ is the complement to the expected total variation distance between $p$ and $q$. Therefore, $\alpha$ describes exactly how well $p$ approximates $q$. If the generation cost coefficient $c_e = \frac{M_p}{M_q}$ is known, where $M_p$ and $M_q$ represent the time it takes for $P$ and $Q$ to generate one full-length sequence, the wall-time speedup for drafts of length $\gamma$ can be computed as:

$$\frac{1 - \alpha^{\gamma+1}}{(1 - \alpha)(\gamma c_e + 1)}. \tag{1}$$

Therefore, an effective speculative decoder should aim to maximize $\alpha$ or minimize $c_e$. Importantly, the set of tokens that increase $\alpha$ is not limited to the highest likelihood token under the target model. Rather, there exists a set of completions $E$ which satisfy the requirements of Algorithm 1. While speculative decoding excels at proposing any given token from $E$, only a subset of these completions may align with biologically meaningful constraints. For instance, two candidate sequences "EEL" and "VLL" (as in Figure 1) might both pass the maximal coupling acceptance check, yet "VLL" may correspond to a conformation with higher predicted structural stability. Current speculative decoding frameworks lack a mechanism to prioritize such candidates, as they do not account for auxiliary objectives, which are essential in protein design but not explicitly modeled during generation.

## 2.2 K-mers

K-mers are contiguous substrings of length $k$ extracted from biological sequences. In proteins, k-mers often correspond to recurring structures such as alpha-helices or beta-strands, capturing local features that are critical to folding and function. Compared to analogous representations in other domains, protein k-mers encode a dense combination of physicochemical properties, structural motifs, and functional sites, all at minimal computational cost. In contrast, n-grams in natural language primarily reflect syntactic or statistical patterns without necessarily encoding physical or functional constraints. Due to their high information content, k-mers have been used across many applications in computational biology, including tasks such as classification, motif discovery, or alignment-free sequence comparison [18]. Notably, Tranception [19] utilizes k-mer attention to better capture structural dependencies during the generation process.

## 3 Methods

### 3.1 SpecMER: K-mer Guided Protein Generation

SpecMER leverages k-mer statistics to guide protein sequence generation. Figure 1 details the modified sampling procedure of SpecMER. Instead of sampling one completion at a time from

the draft model, SpecMER batch generates multiple completions, which we refer to as candidate sequences. A k-mer scoring function then assesses each length $L$ candidate sequence, providing a score based on the selected values of $k$. The highest-scoring candidate is then scored by the draft model and verified by the target model. To summarize, SpecMER decodes protein sequences using the following steps, which repeat until generation is complete:

1. **Candidate construction.** The draft model first samples $L$ tokens $c$ times, producing a batch with dimensions $c \times L$ of candidate sequences.

2. **K-mer scoring.** A k-mer scoring function then selects the candidate sequence $[\tilde{x}^{(1)}(t + 1), \ldots, \tilde{x}^{(c)}(t + L)]$ which yields the highest score.

3. **Conditional probability computation.** For every token $i < L$, the selected candidate sequence is scored using the draft and target models.

4. **Draft selection.** Each proposed token in the candidate sequence is accepted or rejected in accordance with Algorithm 1. The new context is set to $x(t + i) = \tilde{x}(t + i)$, where $i$ is the number of accepted tokens to add.

### 3.2 K-mer Scoring

After the draft model samples $c$ candidate sequences, each sequence is evaluated based on its k-mer frequency. K-mers are computed prior to generation using a multiple sequence alignment (MSA) constructed for a target wild-type protein sequence. MSA data consists of homologous sequences identified based on similarity to the wild-type protein. For each wild-type sequence, we retrieve its corresponding MSA from ProteinGym [20]. The resulting alignment encapsulates evolutionary and functional constraints, making MSA-derived data particularly well-suited for k-mer analysis.

We extract k-mers by applying a sliding window of size $k$ across all sequences in the MSA. We selected $k$ values of $1, 3$, and $5$, in accordance with [19]. We chose not to exceed $k = 5$ as the number of possible k-mers can grow exponentially relative to the vocabulary size, increasing access cost during inference. K-mers are tracked over all sequences in the MSA as they appear, enabling better memory and computational efficiency during retrieval. The resulting k-mer frequencies serve as a proxy for biological plausibility and are normalized to form a probability distribution. During the decoding process, k-mers are used to score candidate sequences $s$:

$$\text{Score}(s) = \frac{1}{L} \sum_{k \in K} \sum_{i=0}^{L-k} \mathbb{P}_k(s(i : i + k)) , \tag{2}$$

where $L$ is the length of the candidate sequence, $K$ is the set of k-mers to evaluate, and $\mathbb{P}_k$ is the probability of a k-mer given the normalized distribution from the MSA. The scoring is additive rather than multiplicative to avoid a zero score in the case of unseen k-mers and to promote exploration of sequences with partially formed motifs. The candidate with the highest k-mer score is selected to continue the decoding process. A significant advantage of k-mer scoring is that it can be pre-computed and is easy to access during generation, contributing negligible computational overhead. Consequently, SpecMER has the same computational complexity as speculative decoding: $\mathcal{O}(L^2)$.

### 3.3 Performance Gain from K-mers

Vanilla speculative decoding drafts individual sequences and verifies each token in the sequence independently. The maximal coupling acceptance algorithm (Algorithm 1) ensures that selected tokens do not diverge from the target model distribution, increasing generation speed with no sacrifice to the quality of the generated sequences. However, there could be many tokens that are consistent with the target distribution at each verification step, passing the acceptance criteria $\eta \leq \min(1, \frac{Q(x)}{P(x)})$. To quantify the looseness of this criterion and the efficacy of external scoring functions, we describe the conditions for selecting tokens consistent with the target model and optimal under the scoring function.

**Proposition 4.4.** *The expected batch-and-select acceptance satisfies:*

$$\mathbb{E}[A^*] = 1 - (1 - \alpha)^m - \epsilon , \tag{3}$$

*where $\mathbb{E}[A^*]$ is the acceptance ratio of the speculative decoder, $\alpha$ is the acceptance ratio of vanilla speculative decoding, and $m$ is the number of batch generated candidate sequences.*

*Proof.* Let P be the draft model and Q the target as before. For a drafted (candidate) sequence $s$, let $M(s)$ be an indicator of the acceptance or rejection of $s$. That is, $M(s) = 1$ if Q accepts $s$, and $M(s) = 0$ otherwise. For a given context sequence $x$,

$$\mathbb{P}[M(s) = 1 \mid s] = r(s) = \min\left(1, \frac{Q(s|x)}{P(s|x)}\right). \tag{4}$$

If $s$ is a single draw, as in vanilla speculative decoding, then the acceptance rate $\alpha$ is defined as $\mathbb{P}_{s \sim P}[M(s) = 1] = \mathbb{E}_{s \sim P}[r(s)]$. In the case of batch generation, $m$ [1] i.i.d. candidates are selected for each draft: $s_1, \ldots, s_m \sim P$. Following drafting, the scoring function $S$ selects the highest scoring candidate $s^* = \text{argmax}_i S(s_i)$. Let acceptance indicator $A^* = M(s^*)$, and define $E = \{\exists i : M(s_i) = 1\}$ as the event in which at least one $s_i$ is an acceptable candidate under $Q$. The probability that an acceptable candidate $s_i$ exists but is not selected by $S$ is defined by the misranking loss $\epsilon = \mathbb{P}[E \wedge (A^* = 0)]$. Consider the case where no candidates satisfy the acceptance criteria of $Q$, $\mathbb{P}[E^c] = \mathbb{P}[M(s_1), \ldots, M(s_m)] = (1 - \alpha)^m$. Intuitively, $(1 - \alpha)^m$ is the probability that among $m$ candidates, none pass the criteria of $Q$. The probability that a selected candidate will be rejected is then defined by $\mathbb{P}[A^* = 0] = \mathbb{P}[E^c] + \mathbb{P}[E \wedge (A^* = 0)] = (1 - \alpha)^m + \epsilon$. The overall acceptance of the decoder is then:

$$\mathbb{E}[A^*] = 1 - \mathbb{P}[A^* = 0] = 1 - (1 - \alpha)^m - \epsilon. \quad \square \tag{5}$$

In practice, $\mathbb{E}[A^*]$ is measured empirically as the average acceptance ratio of the speculative decoder, and $\alpha$ is the average acceptance ratio of vanilla speculative decoding (no external scoring). $\epsilon$ is the probability that the scoring function chooses the wrong candidate, given that there was an acceptable candidate to choose. A good scoring function should yield $\epsilon \ll 1$, indicating that it excels in selecting candidates that will be accepted by $Q$. Additional bounds on expected speedups for given hardware configurations can be found in Appendix A.

## 4 Experiments

In this section, we detail experiments across seven different proteins. We tested SpecMER with ProGen2 [4], an autoregressive protein language model. For each experiment, we conditioned generation on a fixed context length from a given wild-type protein as detailed in Table 1. We selected proteins with varying molecular functions and lengths to ensure robustness of testing. Each experiment consisted of generating 200 sequences on an NVIDIA RTX A6000 GPU. Sequences were generated up until the length of the wild-type protein or until a stop token was generated. We swept across a set of hyperparameters for each decoding method to determine the configuration with the highest acceptance ratio and generation speed. Results of a full hyperparameter sweep are located in Appendix B.3.

### 4.1 Datasets

We focus on the conditional generation task, where the objective is to generate starting from a context sequence. We selected seven proteins with varying functions and collected their MSA from ProteinGym [20]. For each protein, we set the context length to roughly 10% of the wild-type sequence, balancing exploration of sequence space while avoiding common pitfalls in autoregressive generation such as pathological repetition. Table 1 details each protein and its respective molecular function.

Table 1: Summary of proteins and context length used.

| Protein | Description | Molecular Function | Length | Context | MSA Sequences | Citation |
|---------|-------------|--------------------|--------|---------|---------------|----------|
| GFP | Green fluorescent protein | Fluorescence | 238 | 20 | 396 | [21] |
| RBP1 | RalA-binding protein 1 | Stability | 52 | 10 | 135922 | [22] |
| ParD3 | Antitoxin ParD3 | Growth enrichment | 93 | 15 | 38613 | [23] |
| GB1 | IgG-binding domain of protein G | Binding | 56 | 10 | 44 | [24] |
| Bgl3 | $\beta$-glucosidase | Enzyme function | 501 | 50 | 105913 | [25] |
| ADRB2 | Beta-2 adrenergic receptor (GPCR) | Receptor activity | 413 | 40 | 204722 | [26] |
| CBS | Cystathionine beta-synthase | Growth | 551 | 50 | 19563 | [27] |

---

[1]We opt to use $m$ in place of $c$ to avoid confusion with set complement.

## 4.2 Experimental Setup

**Hyperparameters.** Each experiment consisted of generating 200 protein sequences given the starting context sequence (Table 1). We swept over the following hyperparameters: draft tokens $\gamma \in \{5, 10, 15\}$, temperatures $T \in \{0.7, 1, 1.4\}$, and k-mers $k \in \{(1), (3), (1,3), (1,3,5)\}$. Sequences were sampled using nucleus (top-$p$) sampling, setting $p = 0.95$. We opted for a large value of $p$ to enable diverse sampling, controlling how deterministic the output is instead with temperature.

**Evaluation metrics.** We measured four categories across all experiment configurations: acceptance ratio, negative log-likelihood (NLL), top-20 NLL, and top-5 NLL. Acceptance ratio $\alpha$ is defined as:

$$\alpha = \frac{\text{No. accepted tokens}}{\text{No. accepted tokens + No. rejected tokens}} \ , \tag{6}$$

where a higher acceptance ratio is better. After generation, we computed NLL, top-20 NLL, and top-5 NLL under the target model, normalizing for length. We assessed the top-20 and top-5 NLL, as protein design experiments often create a library of sequences and filter for only the most plausible candidates. Sequences with higher likelihoods are more likely to resemble natural proteins, making likelihood a proxy for sequence quality [28, 29]. In addition to likelihood, we computed predicted local-distance difference test (pLDDT) scores [30] using ESMFold [31]. The pLDDT score measures how confident the structure prediction is for a given protein sequence, and has been demonstrated to correlate with the likelihood that a protein sequence will fold into a stable structure [30]. We evaluated biological plausibility using both sequence likelihood and pLDDT, under the assumption that sequences achieving high scores on both metrics are more likely to adopt stable, protein-like structures consistent with natural biological constraints. Sequence likelihoods and pLDDT scores for each wild-type sequence can be found in Appendix B.3.

We generated embeddings for both MSA and sampled sequences ($c = 1, 2, 3, 5$) using ESM2 [31], and used PCA to qualitatively assess alignment between natural and generated sequences. For each decoding method, we selected the top three configurations by average NLL, further filtering for the 100 sequences with the highest pLDDT scores. Bgl3, ADRB2, and CBS were excluded from the pLDDT score analysis due to sequence length limitations of ESMFold.

Generation speed experiments were conducted using ProGen2-S and ProGen2-M as baselines. We tested vanilla speculative decoding ($c = 1$) and SpecMER ($c = 2, 3, 5$) against these baselines, tracking tokens per second. We swept over a range of hyperparameters to find the fastest configuration, averaging over 20 generated sequences.

## 4.3 Results

**Acceptance and log-likelihood.** The best results per experimental configuration are reported in Table 2. We observed that acceptance ratios were marginally higher on average for SpecMER compared to speculative decoding and increased with $c$. Sequences generated with SpecMER had significantly lower NLL, including a lower top-20 and top-5 NLL. That is, while k-mer scoring increased acceptance for some configurations, it nearly unanimously produced protein sequences with higher likelihood.

We observed that the acceptance ratio was sensitive to $T$ and $k$. Specifically, $k$ strongly influenced the acceptance ratio for some proteins, while others were relatively unaffected. For example, GFP exhibited the best performance with $k = \{1, 3\}$, whereas RBP1 had no preference toward $k$. A full discussion on important hyperparameters can be found in Appendix B.3.

**Generation of plausible sequences.** SpecMER yielded sequences with higher likelihoods compared to speculative decoding, indicating that sequences generated by this framework are more likely to be biologically plausible. To corroborate this claim, we computed pLDDT scores, a measure that correlates with the likelihood that a protein will fold into a stable conformation. Table 3 details the average pLDDT scores across 300 sequences per configuration. We observed that on average, SpecMER produces sequences with higher pLDDT scores. GFP was an exception to this trend, exhibiting higher scores for vanilla speculative decoding. However, top-5 pLDDT scores for GFP demonstrated that top-end performance was comparable between all decoding methods (Table 10 in Appendix D). The pLDDT scores did not increase linearly with $c$; instead, the best value of $c$ varied across proteins. This suggests that the optimal value of $c$ reflects a trade-off between exploration of diverse sequence space and adherence to the likelihood distribution of the target model. Furthermore, this trade-off may depend on structural characteristics of the target protein.

Table 2: Decoding results using ProGen2. Each metric is averaged over 200 generated sequences. The value of $c$ chosen for SpecMER experiments represents the number of batch-generated candidate sequences, with $c = 1$ indicating speculative decoding. For each method, a sweep over hyperparameters is conducted as described in Section 4. The best results in each category are reported. A full summary of results is located in Appendix B.3.

| Decoding Method | Protein | Accept Ratio ↑ | NLL ↓ | Top-20 NLL ↓ | Top-5 NLL ↓ |
|---|---|---|---|---|---|
| Speculative Decoding | GFP | $0.911 \pm 0.029$ | $2.45 \pm 0.42$ | $1.38 \pm 0.32$ | $0.98 \pm 0.11$ |
| | RBP1 | $\mathbf{0.938} \pm 0.043$ | $2.73 \pm 0.19$ | $2.43 \pm 0.19$ | $2.01 \pm 0.44$ |
| | ParD3 | $0.902 \pm 0.036$ | $1.93 \pm 0.59$ | $0.80 \pm 0.21$ | $\mathbf{0.50} \pm 0.09$ |
| | GB1 | $\mathbf{0.927} \pm 0.040$ | $2.79 \pm 0.15$ | $2.60 \pm 0.25$ | $2.28 \pm 0.33$ |
| | Bgl3 | $0.852 \pm 0.025$ | $0.91 \pm 0.11$ | $0.76 \pm 0.06$ | $0.67 \pm 0.08$ |
| | ADRB2 | $0.866 \pm 0.079$ | $1.90 \pm 0.65$ | $1.18 \pm 0.30$ | $0.78 \pm 0.16$ |
| | CBS | $\mathbf{0.910} \pm 0.051$ | $2.42 \pm 0.42$ | $2.06 \pm 0.44$ | $1.43 \pm 0.38$ |
| SpecMER ($c = 3$) | GFP | $0.937 \pm 0.076$ | $1.23 \pm 0.68$ | $0.44 \pm 0.04$ | $0.35 \pm 0.04$ |
| | RBP1 | $0.926 \pm 0.038$ | $2.58 \pm 0.32$ | $2.08 \pm 0.39$ | $1.48 \pm 0.21$ |
| | ParD3 | $0.919 \pm 0.093$ | $1.56 \pm 0.49$ | $0.75 \pm 0.12$ | $0.54 \pm 0.06$ |
| | GB1 | $0.924 \pm 0.043$ | $2.71 \pm 0.18$ | $2.43 \pm 0.26$ | $2.06 \pm 0.57$ |
| | Bgl3 | $\mathbf{0.869} \pm 0.030$ | $0.81 \pm 0.14$ | $0.65 \pm 0.07$ | $0.54 \pm 0.08$ |
| | ADRB2 | $0.857 \pm 0.062$ | $1.33 \pm 0.50$ | $0.77 \pm 0.13$ | $0.61 \pm 0.04$ |
| | CBS | $0.902 \pm 0.081$ | $2.17 \pm 0.66$ | $1.47 \pm 0.50$ | $0.84 \pm 0.16$ |
| SpecMER ($c = 5$) | GFP | $\mathbf{0.945} \pm 0.084$ | $\mathbf{1.09} \pm 0.64$ | $\mathbf{0.41} \pm 0.07$ | $\mathbf{0.33} \pm 0.02$ |
| | RBP1 | $0.926 \pm 0.047$ | $\mathbf{2.41} \pm 0.40$ | $\mathbf{1.72} \pm 0.30$ | $\mathbf{1.25} \pm 0.12$ |
| | ParD3 | $\mathbf{0.942} \pm 0.063$ | $\mathbf{1.33} \pm 0.41$ | $\mathbf{0.67} \pm 0.12$ | $0.52 \pm 0.05$ |
| | GB1 | $0.925 \pm 0.038$ | $\mathbf{2.61} \pm 0.27$ | $\mathbf{2.20} \pm 0.31$ | $\mathbf{1.74} \pm 0.15$ |
| | Bgl3 | $0.867 \pm 0.022$ | $\mathbf{0.80} \pm 0.17$ | $\mathbf{0.63} \pm 0.11$ | $\mathbf{0.46} \pm 0.05$ |
| | ADRB2 | $\mathbf{0.869} \pm 0.082$ | $\mathbf{1.03} \pm 0.60$ | $\mathbf{0.57} \pm 0.11$ | $\mathbf{0.43} \pm 0.04$ |
| | CBS | $0.908 \pm 0.066$ | $\mathbf{1.87} \pm 0.68$ | $\mathbf{1.14} \pm 0.32$ | $\mathbf{0.74} \pm 0.07$ |

Table 3: Average pLDDT scores across four different proteins. Sequences are collected from the three best configurations for each decoding method, determined by the highest average log-likelihood under the target model (ProGen2-M).

| Protein | Speculative Decoding ($c = 1$) | SpecMER ($c = 2$) | SpecMER ($c = 3$) | SpecMER ($c = 5$) |
|---|---|---|---|---|
| GFP (↑) | $\mathbf{0.493} \pm 0.156$ | $0.449 \pm 0.160$ | $0.479 \pm 0.191$ | $0.426 \pm 0.176$ |
| RBP1 (↑) | $0.571 \pm 0.116$ | $0.664 \pm 0.127$ | $0.700 \pm 0.115$ | $\mathbf{0.740} \pm 0.105$ |
| ParD3 (↑) | $0.638 \pm 0.206$ | $\mathbf{0.650} \pm 0.202$ | $0.584 \pm 0.201$ | $0.519 \pm 0.176$ |
| GB1 (↑) | $0.465 \pm 0.084$ | $0.464 \pm 0.071$ | $0.477 \pm 0.075$ | $\mathbf{0.504} \pm 0.09$ |

When compared to speculative decoding ($c = 1$), SpecMER ($c = 5$) generated sequences that more closely aligned with homologous sequences from an MSA, while also exploring beyond the MSA centroid. Figure 2a illustrates this result for RBP1, where SpecMER captures both MSA-like sequences and novel variants. Notably, many of these divergent sequences exhibited high likelihood under the target model and achieved higher pLDDT scores on average (Figure 2 and Table 3), indicating a greater degree of biological plausibility. Beyond structural plausibility, we also assessed sequence diversity and found that both speculative decoding and SpecMER generated novel sequences far from the wild-type while maintaining high inter-sequence diversity (see Appendix D.1). Taken together, these results suggest that k-mer guidance preserves evolutionary plausibility while facilitating exploration of high-quality regions of sequence space.

To investigate the role of MSA-derived k-mers, we conducted two ablation experiments (see Appendix C for full details). First, we tested SpecMER under cross-protein mismatches, conditioning on GFP while using GB1-derived k-mers to select continuations, and conditioning on GB1 while using Bgl3-derived k-mers. In both cases, likelihoods dropped relative to protein-specific k-mers, indicating that observed gains depend on the correct evolutionary context. Second, we reduced the

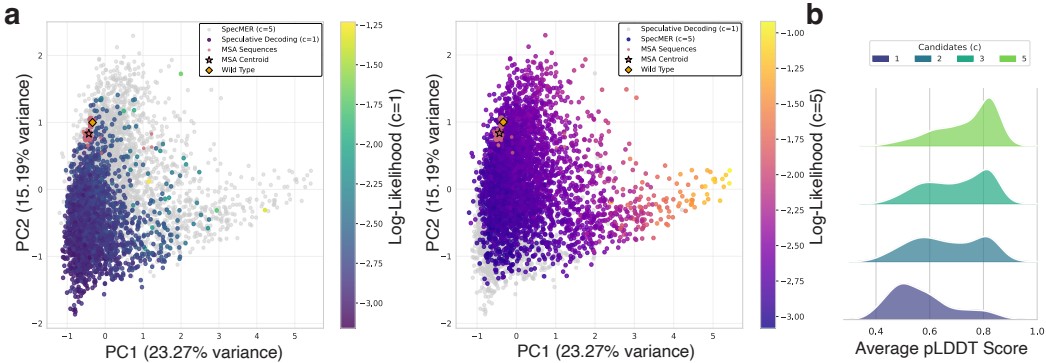

Figure 2: Impact of candidates $c$ on biological plausibility of sequences generated by vanilla speculative decoding ($c = 1$) versus SpecMER for RBP1. **a**, PCA of embeddings from sequences generated by speculative decoding ($c = 1$) (highlighted on the left plot) and SpecMER ($c = 5$) (highlighted on the right plot), compared to embeddings from the MSA of RBP1. Each sequence is shaded by its likelihood under the target model (ProGen2-M). Sequences generated by SpecMER cluster more closely to the MSA embeddings and demonstrate a higher likelihood, indicating a higher degree of biological plausibility. **b**, pLDDT distributions for different configurations of SpecMER, with $c = 1$ representing vanilla speculative decoding. SpecMER demonstrates higher structural confidence scores, indicating that generated sequences are more likely to fold.

depth of the Bgl3 MSA from ~105k to 1k sequences. The resulting likelihoods steeply declined, supporting our conclusion that MSA quality impacts the performance of SpecMER.

While vanilla speculative decoding ($c = 1$) routinely undersamples the high-likelihood tail of the target distribution (Figure 1c; see Section 2.1), our experiments demonstrate that SpecMER overcomes this limitation. By using k-mers extracted from the MSA to bias the draft model toward structurally grounded completions, SpecMER achieves improved coverage of biologically plausible sequences, often exceeding the likelihoods obtained by target-only decoding. Table 4 compares the top-20 NLL values from SpecMER ($c = 5$) compared to those from the target model, highlighting the improved coverage of high-likelihood sequences.

Table 4: Comparison of top-20 NLL from sequences generated by the target model (ProGen2-M) versus SpecMER using the same temperature values. Sequences are scored using ProGen2-M.

| Method | Bgl3 | GFP | RBP1 | GB1 | ParD3 |
|---|---|---|---|---|---|
| Target | $0.78 \pm 0.02$ | $0.51 \pm 0.04$ | $1.62 \pm 0.15$ | $2.27 \pm 0.24$ | $0.69 \pm 0.11$ |
| SpecMER ($c = 5$) | $0.63 \pm 0.11$ | $0.41 \pm 0.07$ | $1.72 \pm 0.30$ | $2.20 \pm 0.31$ | $0.67 \pm 0.12$ |

**Wall-time speedups.** We observed an increase in sequence likelihoods as the number of candidates increased (Table 2a). This improvement, however, incurred a slight cost to generation speed, indicating a trade-off between sequence quality and computational efficiency. As demonstrated in Table 5, generating more candidates slowed decoding, though the decline in speed was minimal. Speculative decoding ($c = 1$) achieved the fastest generation speed with an average speedup of $32\%$, while SpecMER ($c = 5$) yielded the slowest configuration, achieving a modest $24\%$ speedup. Notably, $c = 3$ yielded the largest increase in sequence log-likelihood for the smallest decrease in speed. We noticed an increase in variance as $c$ increased due to batch generation, which we discuss further in Appendix B.1.

We also tested the generation speed with ProGen2-XL as the target model and ProGen2-M as the draft, finding that with $c = 3$, SpecMER demonstrated a speed increase of $38\%$ over target-only decoding. In reference to the example from the introduction, SpecMER ($c = 3$) would have taken 40 hours to generate 20,000 protein sequences of length 200 amino acids, saving an entire day of computing.

Table 5: Generation speed (tokens/sec) over the best configuration per method. Results are averaged over GFP, RBP1, and GB1. The draft and target models are ProGen2-S and ProGen2-M, and the baseline is speculative decoding ($c = 1$). Speedup is calculated with respect to the target model.

| - | **Draft** | **Target** | **Baseline** | **SpecMER** ($c = 2$) | **SpecMER** ($c = 3$) | **SpecMER** ($c = 5$) |
|---|---|---|---|---|---|---|
| Toks/sec | 74.11 | 31.48 | $41.62 \pm 0.88$ | $41.43 \pm 1.24$ | $40.57 \pm 2.83$ | $39.11 \pm 4.50$ |
| Speedup | - | - | 32% | 32% | 29% | 24% |

While increasing the number of candidates decreased tokens per second, it resulted in selecting better candidates the majority of the time. SpecMER ($c = 5$) had the lowest misranking error (Equation 3 and Figure 3), selecting an acceptable sequence from a pool of candidates 92% of the time. Of the set of tokens consistent with the target distribution, SpecMER was able to correctly discern between tokens that increased acceptance and tokens that increased both acceptance and biological plausibility. To summarize the trade-off space, increasing $c$ decreases $\epsilon$, tokens per second, and negative log-likelihood, while increasing acceptance ratio and pLDDT scores. Further discussion on this trade-off space can be found in Appendix B.2.

## 5   Conclusion

**Discussion.** In this study, we developed the first speculative decoding framework for protein generation. Using ProGen2-S as the draft model and ProGen2-M as the target model, we achieved an average speedup of 32% across functionally diverse proteins. Building on this foundation, we developed SpecMER, a novel framework that incorporates k-mer guidance derived from multiple sequence alignments to guide generation toward biologically meaningful regions of sequence space.

SpecMER drafts multiple candidate sequences and uses evolutionary patterns from k-mers to select completions that are more likely to fold into stable, biologically plausible proteins. As illustrated in Figure 1, speculative decoding often undersamples high-likelihood sequences, despite proposing tokens consistent with the target model distribution. SpecMER addresses this deficiency by guiding generation toward sequences that both align with MSA-derived motifs and exhibit improved log-likelihood and pLDDT scores. In doing so, it preserves the speed advantages of speculative decoding while producing sequences that are more likely to fold into stable conformations, achieving the best of both efficiency and quality.

Beyond empirical gains, we also provide bounds on wall-time speedups and define the criteria for selecting tokens that conform to guidance and acceptance objectives. Furthermore, we offer practical guidance for tailoring hyperparameter configurations to maximize performance improvements for protein design workflows. As speculative decoding techniques continue to evolve and more protein data becomes available, we anticipate larger speedups in automated protein generation. SpecMER marks a step forward in accelerating protein design, paving the way for high-throughput design workflows and beyond.

**Limitations.** The effectiveness of SpecMER depends on the quality of the input alignment (MSA); performance may degrade when informative motifs are sparse or unavailable. For example, target proteins with extensive disordered regions or lacking discernible structural patterns may benefit less from k-mer guidance. While k-mers impose minimal computational overhead, increasing the number of drafted candidates $c$ increases compute and therefore energy cost, which may limit scalability in certain settings. SpecMER utilizes batch generation to efficiently draft multiple candidates, though this is not strictly parallel. In practice, a fully parallel implementation could yield even greater speedups, but was not attainable given the limitations of our hardware setup.

**Future work.** While our method was developed for conditional generation of protein sequences, its principles could extend to natural language, where low-cost scoring functions can guide generation under constraints such as style or safety.

## Acknowledgments and Disclosure of Funding

This work was supported by the Parker H. Petit Institute for Bioengineering and Biosciences (IBB) interdisciplinary seed grant, the Institute of Matter and Systems (IMS) seed grant, the National Science Foundation (NSF) Graduate Research Fellowship Program (GRFP), and Georgia Institute of Technology start-up funds.

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

# Appendix

## A Bounding Speedups

**Definition A.1.** Let $c_e$ be the cost coefficient for one iteration of SpecMER. Then,

$$c_e = \frac{cM_p + M_k}{M_q}, \tag{7}$$

where $M_k$ is the time it takes the k-mer scoring function to score $c$ candidates. Assuming $M_p$ can batch generate candidates, then the cost of $M_p$ grows sublinearly with respect to $c$. Additionally, $M_k \ll M_p$, and it follows that:

$$c_e = \frac{\xi M_p}{M_q}, \tag{8}$$

where $1 \leq \xi < c$ is the batch generation cost. Under true parallelism, $\xi = 1$ and the expected improvement factor remains the same as in Equation (1).

**Proposition A.2.** *The expected (batch) wall-time speedup factor of batch generation is*

$$S(\gamma) \approx \frac{1 - \alpha^{\gamma+1}}{(1 - \alpha)[c_e + 1]}. \tag{9}$$

*Proof.* Let $T_{\text{iteration}}(\gamma)$ be the time it takes for one iteration of draft selection and evaluation to complete on $\gamma$ draft tokens. Similarly, let $T_p(\gamma)$, $T_q(\gamma)$, and $T_k$ be the time it takes for the draft model (single candidate), target model, and k-mer scoring function to complete one iteration.

Since $T_p(\gamma)$ can batch generate, $T_p(\gamma) = \xi T_p(\gamma)$, whereas in the serial case, $T_p(\gamma) = cT_p(\gamma)$. The total time for an iteration is then $T_{\text{iteration}}(\gamma) = [\xi T_p(\gamma) + T_k] + T_q(\gamma)$, with an expected speedup of:

$$S(\gamma) = \frac{1 - \alpha^{\gamma+1}}{(1 - \alpha)\left(\frac{T_{\text{iteration}}(\gamma)}{T_q(\gamma)}\right)}. \tag{10}$$

Let us define the cost coefficient $c_{\text{draft}}(\gamma) = \frac{\xi T_p(\gamma) + T_k}{T_q(\gamma)}$. By Equation (8), $c_{\text{draft}}(\gamma) \approx c_e$, simplifying the ratio between total iteration time and target iteration time $\frac{T_{\text{iteration}}(\gamma)}{T_q(\gamma)} \approx c_e + 1$. Consequently,

$$S(\gamma) \approx \frac{1 - \alpha^{\gamma+1}}{(1 - \alpha)[c_e + 1]}. \tag{11}$$

In the case that generation is serial, the wall-time speedup results in a similar equation.

**Corollary A.3.** *The expected serial wall-time speedup factor is:*

$$S(\gamma) \approx \frac{1 - \alpha^{\gamma+1}}{(1 - \alpha)[\frac{c}{\xi}c_e + 1]}. \tag{12}$$

We note that $T_p(\gamma)$ and $T_q(\gamma)$ are functions of $\gamma$ as the time it takes to decode a sequence of length $\gamma$ does not scale linearly. In practice, generating short sequences approximately scales linearly with $\gamma$, in which near exact equality is achieved in Equation (9) and Equation (12). Increasing the acceptance ratio leads to large gains in time saved across all configurations of $\gamma$.

## B Hyperparameters

### B.1 Implementation Details

Code for SpecMER is publicly available here: `https://github.com/amirgroup-codes/SpecMER.git`.

ProGen2 [4] is a decoder-only Transformer architecture with four different parameter sizes: small - 151M; medium/base - 764M; large - 2.7B; xlarge - 6.4B. No model training is performed in this work; inference is run on a server with eight NVIDIA A6000 GPUs.

We conducted extensive testing to find the correct parallel implementation of candidate sequence generation. Batch generation, while not exactly parallel, yielded the best results in regard to tokens per second. We estimate that batch generation took about $1.2 - 1.3\times$ longer to generate $c = 5$ sequences as compared to $c = 1$ sequences, incurring additional latency due to larger batch operations. This discrepancy in generation time resulted in a larger variance in tokens per second as $c$ increased. When a speculated token is rejected, decoding takes longer, and the added cost of batch generation appears in the variance. Due to the non-deterministic nature of sampling, some generations have higher acceptance rates than others; therefore, occasional rises and dips in acceptance have a larger impact on tokens per second than producing a single sequence at a time. We implemented SpecMER on top of the sample method implementation provided by ProGen2. We considered attempting a fully parallel implementation of batched sequence generation; however, we ultimately decided to defer to the default implementation. This was due to hardware limitations and practical considerations. A fully parallel multi-GPU system, while likely to yield a higher tokens per second, is significantly more demanding for practitioners who wish to use SpecMER. We implemented SpecMER with single GPU instances in mind.

In addition to testing configurations for parallel implementations, we tested KV caching versus full rescoring during each iteration of SpecMER. In a normal setting, KV caching would almost always be preferable. However, due to the maximal coupling acceptance logic, full rescoring versus KV caching depended on the expected performance of the decoder. We stored KV caches for the draft model and target model, updating at each iteration; however, if any token was rejected and resampled using the logic in Algorithm 1, that token would require a full forward pass. Since correction tokens are resampled from the residual distribution, the corresponding logits must be recomputed with a full forward pass. This poses a tradeoff: either compute a full uncached forward pass for both the draft and target model at each iteration, or compute the cached forward pass at the risk of incurring an extra forward pass for every rejected token. We observed that as long as the speculative decoder achieved an acceptance ratio of $\approx 80\%$ or higher, the KV cache implementation was faster. For this reason, we chose to use a custom implementation of KV caching for SpecMER. We recognize that our implementation of SpecMER is not fully optimized and could be improved to realize even faster speedups. We leave this task for future work, or for specialized implementations with access to better computing hardware.

## B.2 Trade-off Space

Selecting the number of candidates to draft $c$ comes at the cost of a trade-off between generation speed and generation quality. Figure 3 details how $c$ influences generation speed and misranking error $\epsilon$. Setting $c = 3$ yielded the best tradeoff in this space, demonstrating the largest gain in likelihood improvements for the least decrease in tokens per second. We observed an improvement in misranking error for SpecMER, indicating that structural cues from SpecMER not only increase structural plausibility but also help close the distribution gap between the target and draft model. The target model in protein generation often exhibits higher correlation with functional and structural benchmarks, learning these auxiliary objectives implicitly from millions of protein sequences. This correlation increases as a function of the number of parameters. SpecMER illustrates that this auxiliary gap between the target and draft can be supplanted with external functional or structural information.

## B.3 Results From Hyperparameter Sweep

This section details how each hyperparameter impacted acceptance and likelihood. As stated in the main text, we swept over the following hyperparameters: draft tokens ($\gamma$), temperature ($T$), and k-mers ($k$). We tested every combination of the following configurations: $\gamma \in \{5, 10, 15\}$, $T \in \{0.7, 1, 1.4\}$, $k \in \{(1), (3), (1, 3), (1, 3, 5)\}$. We chose this consolidated list to encompass realistic hyperparameters used for protein generation. For each configuration, we generated 200 protein sequences, stopping generation at max sequence length (length of the wild-type for the context being used) or upon generation of a stop token ("2" in the case of ProGen2). Depending on the protein selected, testing a single configuration could last anywhere between $2 - 30$ minutes to generate 200 full-length sequences. The final hyperparameter set used to report results in Table 2 is listed in Table 6.

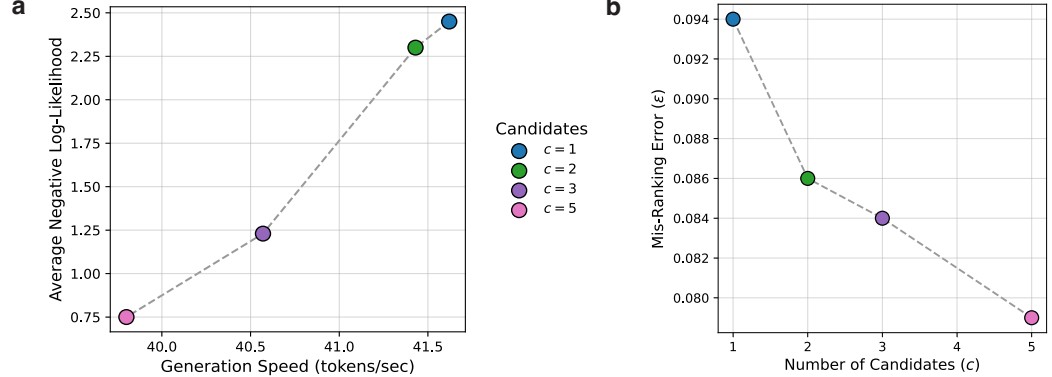

Figure 3: Trade-off space between number of candidates, misranking error, and negative log-likelihood (NLL). **a**, Generation speed versus NLL. As $c$ increases, NLL drastically increases at the penalty of reduced generation speed. **b**, Number of candidates versus misranking error $\epsilon$. As $c$ increases, misranking error decreases, indicating the influence of k-mers on selecting candidates that maximize acceptance while also maximizing the k-mer score.

Table 6: Final hyperparameter configurations for each protein corresponding to results in Table 2.

| Protein | Temperature | Draft Tokens | $k$ value | Candidates |
|---------|-------------|--------------|-----------|------------|
| Bgl3 | 1.0 | 5 | 3 | 5 |
| GFP | 0.7 | 5 | 1, 3 | 5 |
| RBP1 | 1.0 | 10 | 3 | 5 |
| GB1 | 1.4 | 10 | 1, 3, 5 | 5 |
| ParD3 | 1.0 | 5 | 1, 3, 5 | 5 |
| CBS | 0.7 | 5 | 1, 3, 5 | 5 |
| ADRB2 | 0.7 | 5 | 1, 3 | 5 |

We observed variance in generated sequence lengths, largely dependent on temperature. Sharper temperatures $(T = 0.7)$ led to longer sequences on average, shifting probability mass away from producing a stop token early. Sequence likelihoods were heavily influenced by $T$, with likelihood increasing on average when temperature decreased. This is not an unexpected result: decreasing temperature with top-$p$ sampling results in a more deterministic sampling procedure. With this in mind, we chose to generate sequences over a range of temperature values, as increased sequence likelihood does not always result in more plausible structures.

We noticed that each protein had subtle preferences toward the selection of $\gamma$. Shorter proteins, such as ParD3 and GB1, preferred $\gamma = 5$, while longer proteins, such as GFP, saw improvements with $\gamma = 15$. Bgl3 and RBP1 showed no significant preference toward this hyperparameter. Plots pertaining to the effect of $T, k$, and $c$ are available at the conclusion of the appendix.

For additional context to the main results, we include the pLDDT and NLL measurements for each protein in Table 7.

Table 7: Negative log-likelihood (NLL) and pLDDT scores for each wild-type sequence. Missing values indicate proteins for which structural predictions were not available.

| Protein | NLL | pLDDT |
|---------|-----|-------|
| CBS | 0.75 | – |
| Bgl3 | 0.92 | – |
| ADRB2 | 1.31 | – |
| ParD3 | 2.11 | 0.79 |
| GB1 | 2.52 | 0.82 |
| RBP1 | 2.63 | 0.83 |
| GFP | 2.93 | 0.42 |

# C MSA Ablations

We performed two ablation studies to test the efficacy of k-mers in SpecMER.

**Cross-protein k-mers.** We tested SpecMER using mismatched evolutionary signals: (1) generate conditioned on GFP with GB1-derived k-mers, and (2) generate conditioned on GB1 with Bgl3-derived k-mers. The results are shown in Table 8. In both cases, we observed a drop in likelihoods, both on average and for the top-20 highest-likelihood sequences, compared to the protein-specific results in Table 4. This suggests that protein-specific k-mers are responsible for the increase in likelihood observed with SpecMER.

Table 8: Cross-protein k-mer ablation results.

| Condition | Mean NLL | Top-20 NLL |
|---|---|---|
| GFP + GB1 k-mers | $2.52 \pm 0.27$ | $1.78 \pm 0.23$ |
| GB1 + Bgl3 k-mers | $2.79 \pm 0.10$ | $2.59 \pm 0.11$ |

**MSA depth.** We next tested the importance of MSA depth by generating Bgl3 proteins using only 1,000 sequences from the MSA instead of the full set of $\sim$130k sequences. This reduction limited the number of k-mers available to guide generation. The top-20 sequences yielded an average NLL of $1.56 \pm 0.20$, compared to $0.63 \pm 0.11$ for SpecMER ($c = 5$) with the full-depth MSA. These results support our conclusion that SpecMER's performance can degrade when informative motifs are sparse or unavailable.

# D Structural Scores and Embeddings

This section describes experiments pertaining to structural scores, diversity metrics, and embeddings for generated protein sequences. For each section, metrics are computed over all tested configurations of vanilla speculative decoding and SpecMER.

## D.1 Sequence Diversity

Novelty is a key metric for determining how well generative models sample from the space of learned protein sequences. We computed two diversity metrics to assess how SpecMER affects sequence novelty: WT Hamming distance and inter-sequence Hamming distance. WT Hamming distance illustrates how many edits away the generated protein is from the wild-type. Inter-sequence illustrates how similar sequences generated under the same configuration are. Results for both metrics are reported in Table D.1.

Table 9: Wild-type (WT) distance and inter-sequence (Inter-Seq) distance across proteins. SpecMER generates sequences further from the WT while maintaining inter-sequence diversity, thereby exploring sequence space while producing plausible protein sequences.

| Protein | WT Dist. (SpecMER) | WT Dist. (Spec. Dec.) | Inter-Seq (SpecMER) | Inter-Seq (Spec. Dec.) |
|---|---|---|---|---|
| GFP | $208.35 \pm 5.76$ | $208.49 \pm 4.91$ | $181.78 \pm 27.14$ | $184.56 \pm 27.14$ |
| RBP1 | $41.27 \pm 3.48$ | $42.81 \pm 3.43$ | $42.60 \pm 3.87$ | $44.88 \pm 4.00$ |
| ParD3 | $75.97 \pm 3.01$ | $78.68 \pm 2.41$ | $67.39 \pm 6.87$ | $70.00 \pm 5.41$ |
| GB1 | $44.70 \pm 3.25$ | $45.27 \pm 3.27$ | $46.47 \pm 4.46$ | $46.99 \pm 3.86$ |
| Bgl3 | $324.88 \pm 30.88$ | $333.12 \pm 33.96$ | $261.02 \pm 42.28$ | $284.97 \pm 40.44$ |
| CBS | $378.64 \pm 140.60$ | $431.00 \pm 106.05$ | $291.84 \pm 161.60$ | $457.27 \pm 40.98$ |
| ADRB2 | $263.80 \pm 120.22$ | $290.18 \pm 109.01$ | $270.82 \pm 93.46$ | $340.28 \pm 47.66$ |

We observed that SpecMER followed the same trends as speculative decoding for both WT and inter-sequence distance. Both methods generate many dissimilar designs far away from the WT, ensuring that sequence generation explores novel sequence space, a key advantage of autoregressive PLM generation.

## D.2 Embeddings

We assessed embeddings of sequences generated from vanilla speculative decoding and SpecMER. Embeddings were generated using ESM-2 8M [31] with mean pooling and unaligned sequences. ESM-2 is a protein language model that embeds sequences that are functionally or structurally similar closer together. For each protein used as context for conditional generation, we computed embeddings for its respective MSA. Sequences from the MSA are expressed in nature, providing an anchor point for which sequences are likely to be biologically plausible. We used PCA to compare embeddings over two principal components, finding that sequences generated by SpecMER were closer in proximity to the MSA sequences than vanilla speculative decoding. We deduced from this analysis that k-mers formed from the MSA did guide generation toward biologically plausible sequences. However, this analysis alone does not provide enough evidence to support this claim, as embedding distance and PCA are approximate metrics for sequence plausibility. To further corroborate claims of plausibility, we computed structural confidence scores as detailed in the next section. Embedding plots for each protein can be found at the conclusion of the appendix.

## D.3 pLDDT Scores

We assessed pLDDT [30] scores as a structural confidence measure. pLDDT ranges from $0 - 100$ at each amino acid site, with $0$ indicating a structure very unlikely to fold, and $100$ indicating a structure very likely to fold. We computed pLDDT scores using ESMFold [31], a structure prediction model that provides per-residue pLDDT scores. To assess the overall structural confidence of generated proteins, we averaged the pLDDT scores across all amino acid positions. Figure 2b details pLDDT scores from top configurations of vanilla speculative decoding and SpecMER for different $c$ values, with generation conditioned on RBP1. For each candidate $c$, we identified the three best configurations for each method, determined by average log-likelihood. Furthermore, we selected the top-100 most likely sequences from these three configurations, resulting in a set of $300$ data points for each $c$ value. This was done to ensure diversity in the set of sampled sequences. We observed that as $c$ increased, so did the average pLDDT score. This indicates a clear relationship between increasing the influence of k-mers and generating sequences with higher biological plausibility.

We computed average (Table 3) pLDDT scores for all proteins with the exception of Bgl3. This is due to the sequence length, which exceeds 400, and is therefore incompatible with ESMFold. For the remaining proteins, we selected the top 300 sequences across the three best configurations for each method as described before. We observed that sequences generated using SpecMER yielded the highest pLDDT scores, with the exception of GFP, which demonstrated the highest pLDDT scores under vanilla speculative decoding. This was a surprising observation, as conditional generation using GFP as the context sequence yielded higher log-likelihoods when using SpecMER. However, top-5 statistics indicated higher-end scores were comparable for GFP. A summary of top-5 pLDDT scores can be found in Table 10. We observed a similar trend with top-5 and average pLDDT scores. That is, increasing $c$ led to higher pLDDT scores, with the exception of GFP, which was competitive across all values of $c$.

Table 10: Top-5 pLDDT scores across four different proteins. Sequences are collected from the three best configurations for each decoding method, determined by the highest average log-likelihood under the target model (ProGen2-M).

| Protein | Speculative Decoding ($c = 1$) | SpecMER ($c = 2$) | SpecMER ($c = 3$) | SpecMER ($c = 5$) |
|---|---|---|---|---|
| GFP | $0.892 \pm 0.020$ | $0.838 \pm 0.062$ | $0.888 \pm 0.008$ | $0.850 \pm 0.035$ |
| RBP1 | $0.864 \pm 0.019$ | $0.884 \pm 0.016$ | $0.876 \pm 0.016$ | $0.888 \pm 0.011$ |
| ParD3 | $0.912 \pm 0.004$ | $0.918 \pm 0.008$ | $0.922 \pm 0.004$ | $0.926 \pm 0.005$ |
| GB1 | $0.790 \pm 0.030$ | $0.704 \pm 0.010$ | $0.760 \pm 0.059$ | $0.794 \pm 0.019$ |

# E   Multiple Sequence Alignment and K-mer Generation

We collect MSAs of target proteins from ProteinGym [20]. As described in Section 3.2, k-mers are formed by applying a $k$ length sliding window over all sequences in the MSA of a respective protein. Gap characters "-" are ignored.

We observed that the best $k$ value for each protein varied substantially. Some proteins demonstrated little to no preference toward $k$, such as RBP1 and GB1, whereas GFP was heavily influenced by $k = \{1, 3\}$ and $k = \{1, 3, 5\}$. It is possible that for proteins comprised of large disordered regions, SpecMER would show reduced effectiveness, or even be detrimental to generation. Plots describing how k values affected log-likelihood can be found at the conclusion of the appendix.

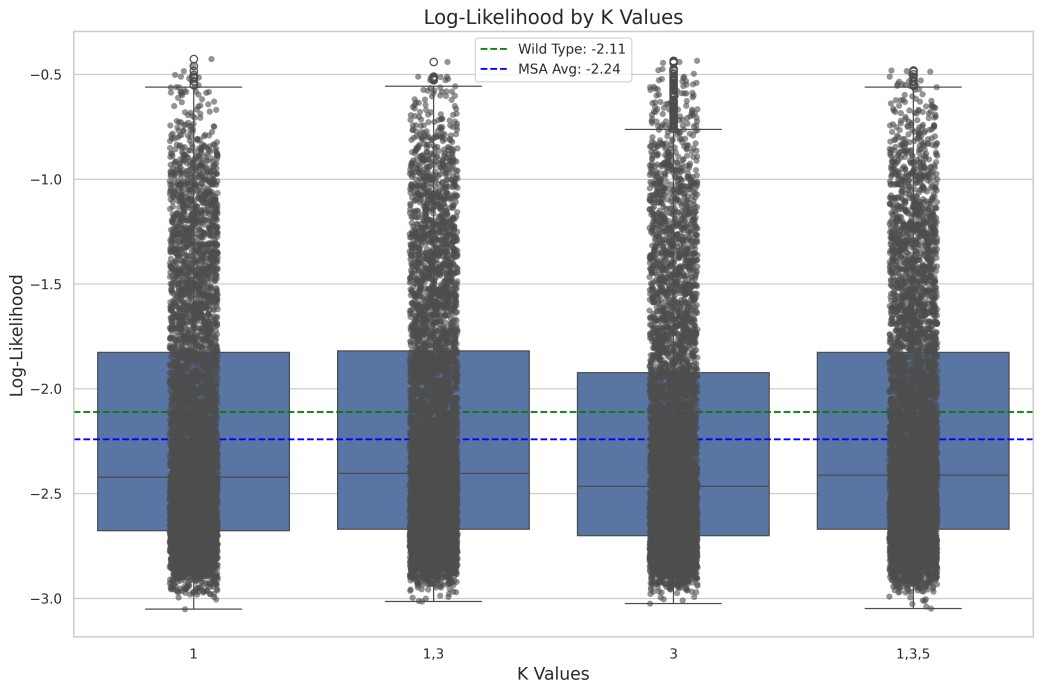

Figure 4: Log-likelihood versus selection of $k$ for ParD3.

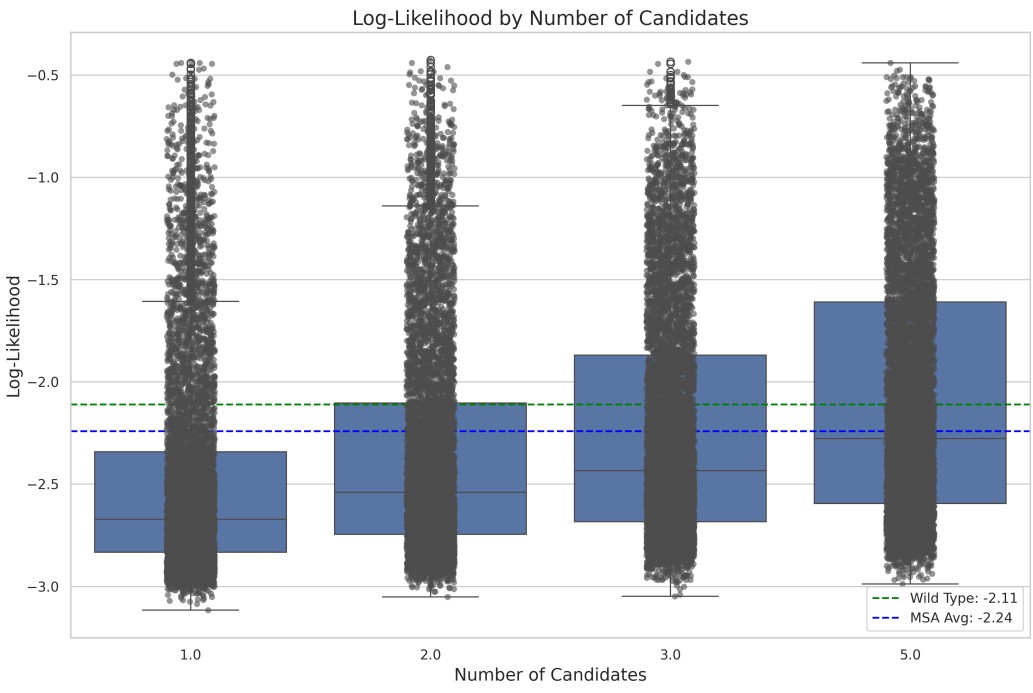

Figure 5: Log-likelihood versus selection of candidates $c$ for ParD3.

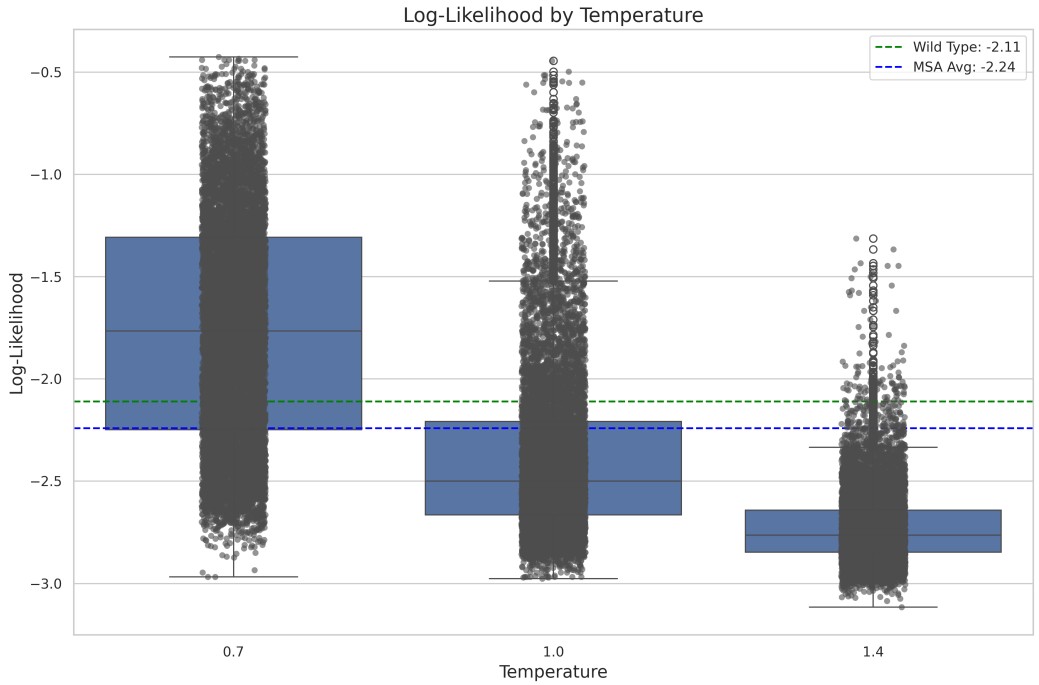

Figure 6: Log-likelihood versus selection of temperature for ParD3.

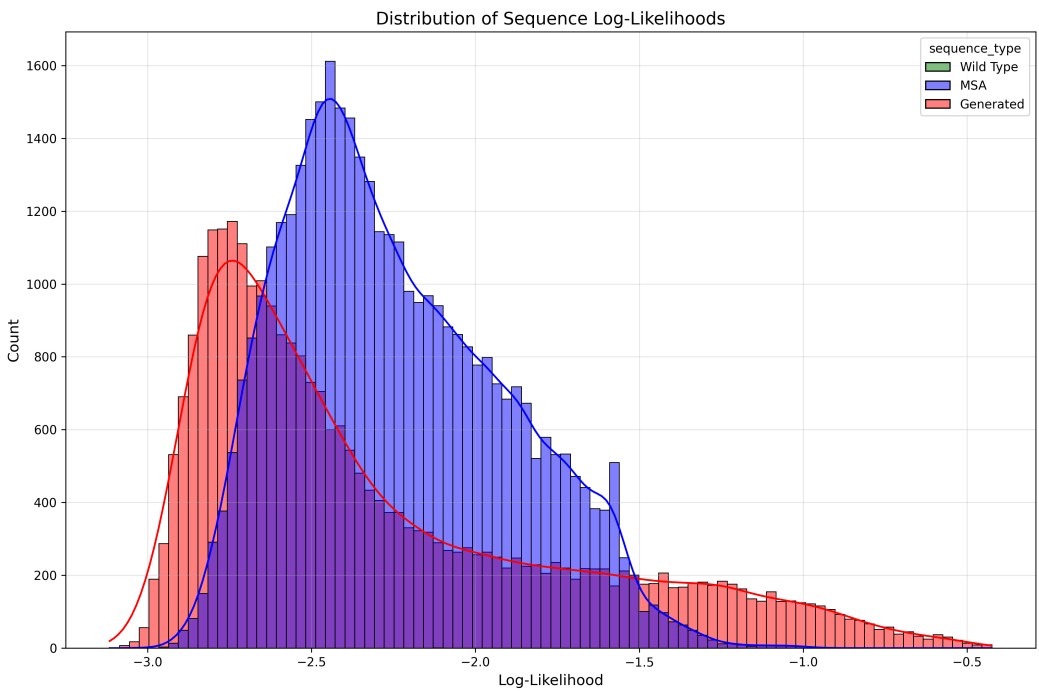

Figure 7: Likelihood distribution of ParD3 sequences generated using SpecMER compared to the likelihood distribution of its collected MSA (scored using ProGen2-M).

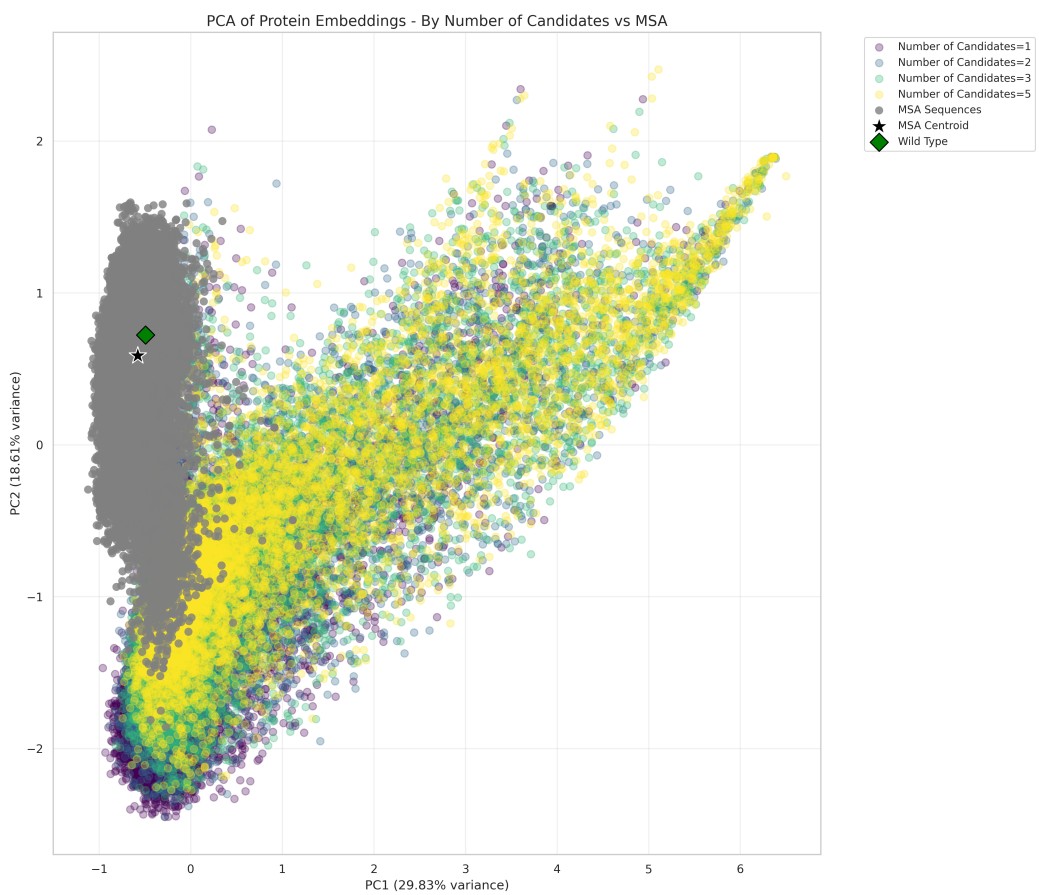

Figure 8: PCA plot for sequence embeddings generated with varying $c$, with $c = 1$ denoting speculative decoding, and $c > 1$ denoting SpecMER. Sequences are generated using the context of ParD3.

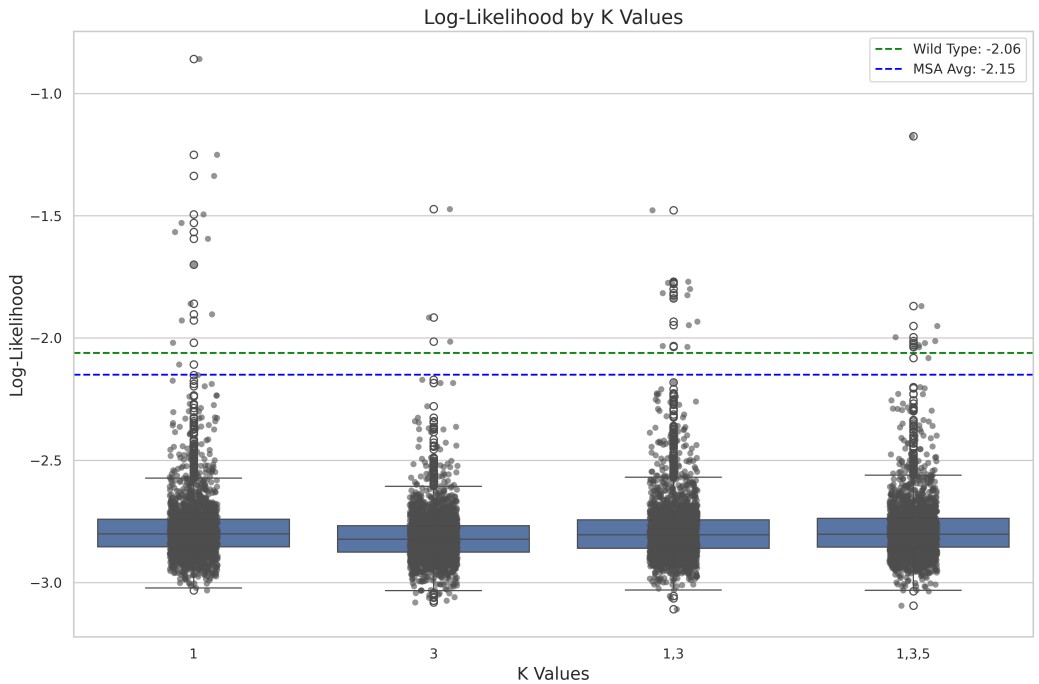

Figure 9: Log-likelihood versus selection of $k$ for GB1.

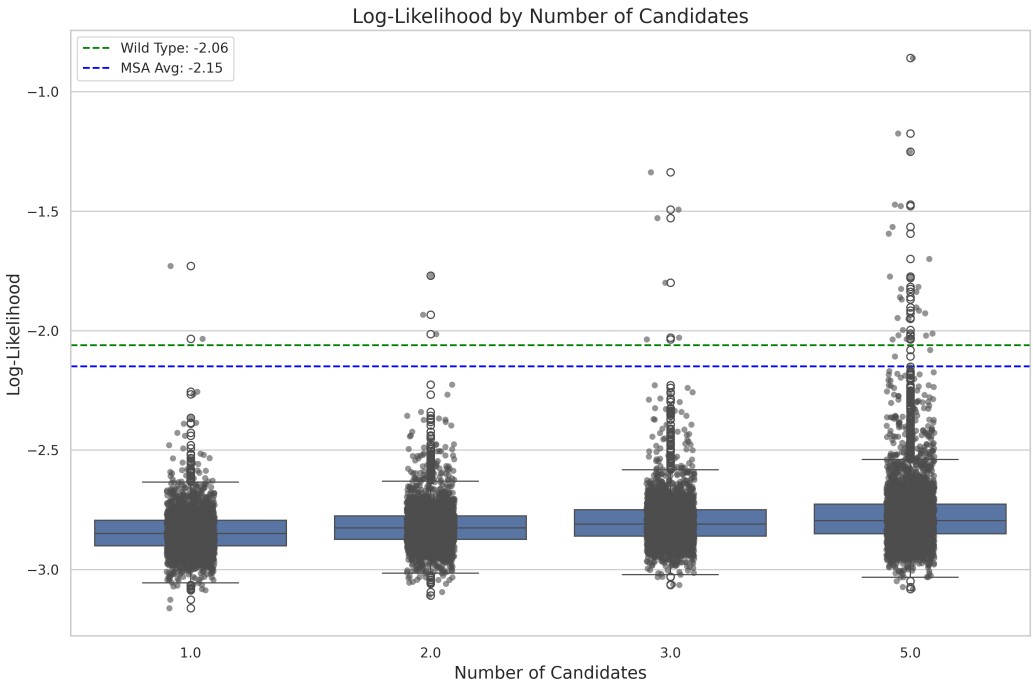

Figure 10: Log-likelihood versus selection of candidates $c$ for GB1.

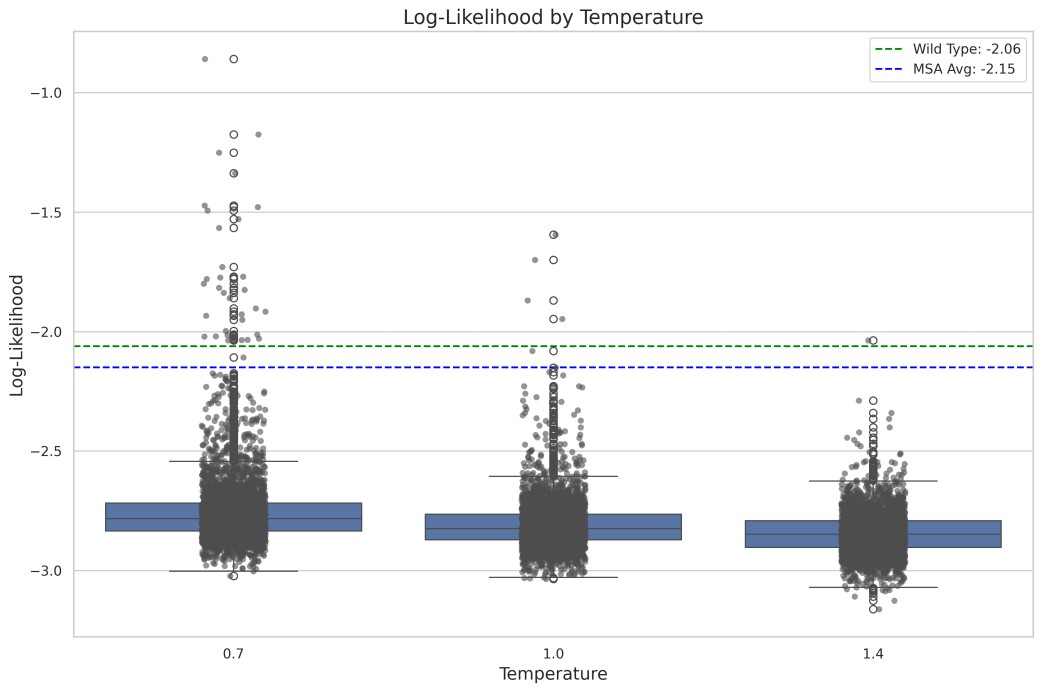

Figure 11: Log-likelihood versus selection of temperature for GB1.

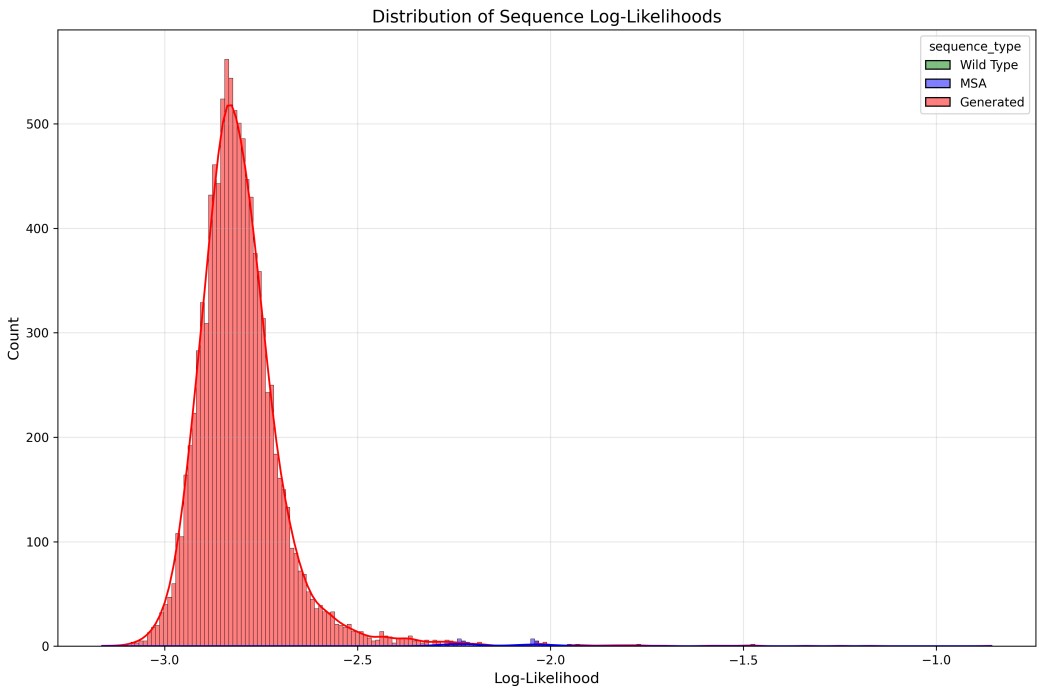

Figure 12: Likelihood distribution of GB1 sequences generated using SpecMER compared to the likelihood distribution of its collected MSA (scored using ProGen2-M).

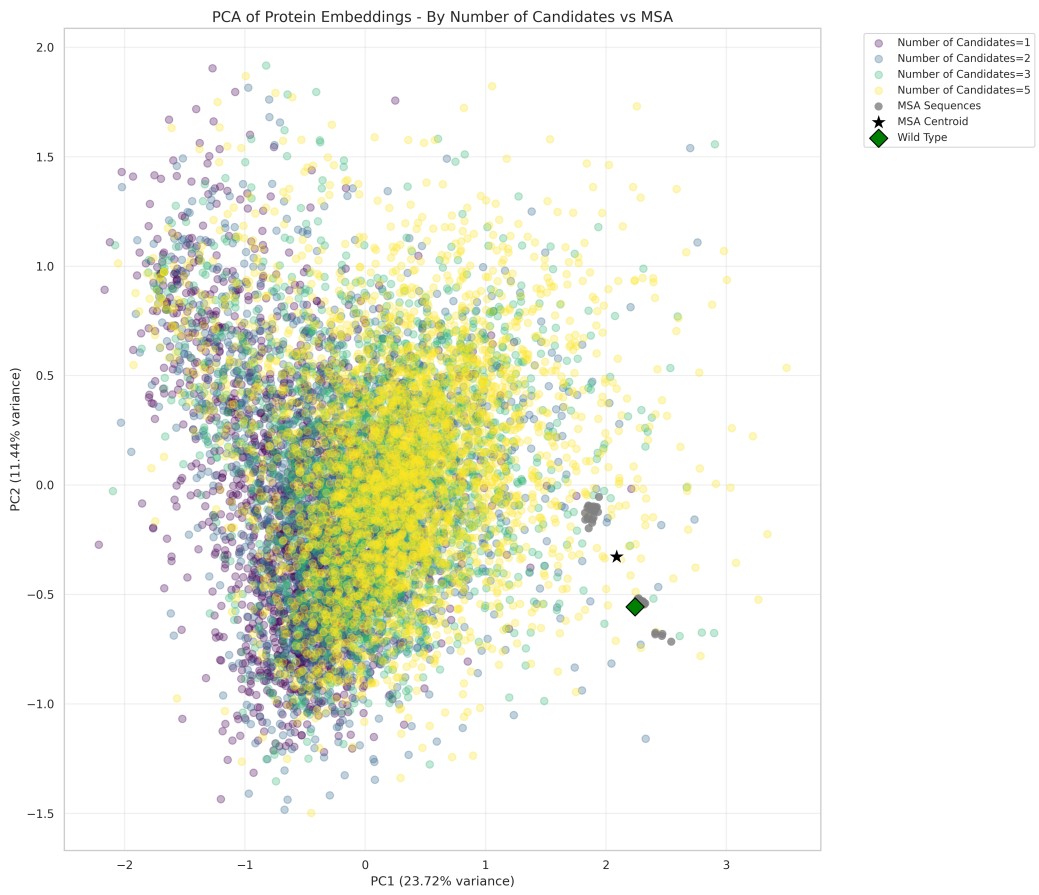

Figure 13: PCA plot for sequence embeddings generated with varying $c$, with $c = 1$ denoting speculative decoding, and $c > 1$ denoting SpecMER. Sequences are generated using the context of GB1.

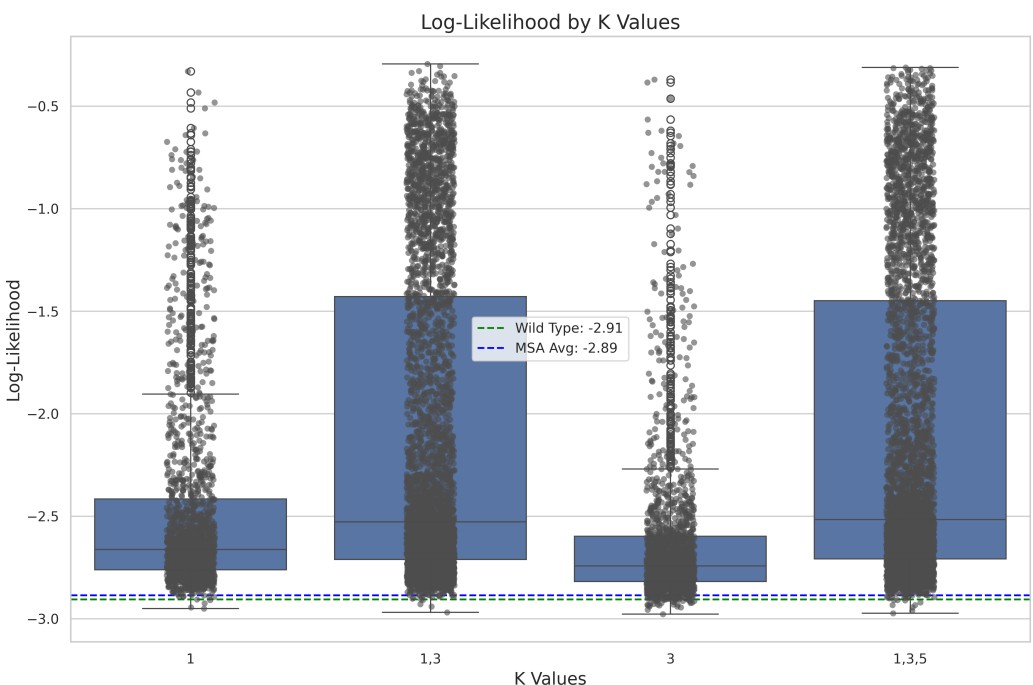

Figure 14: Log-likelihood versus selection of $k$ for GFP.

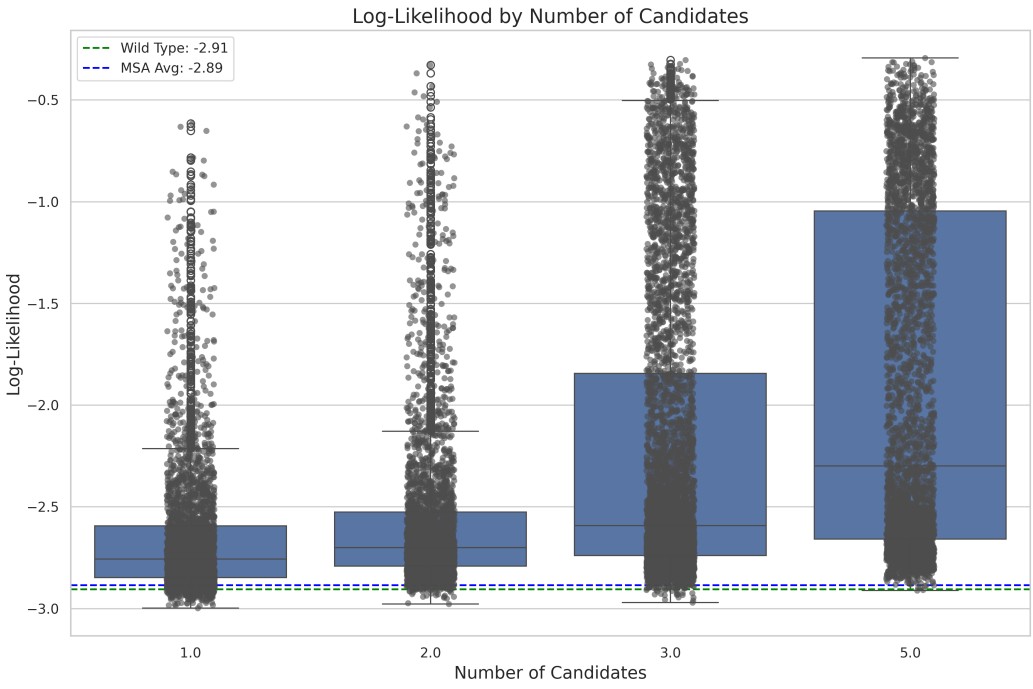

Figure 15: Log-likelihood versus selection of candidates $c$ for GFP.

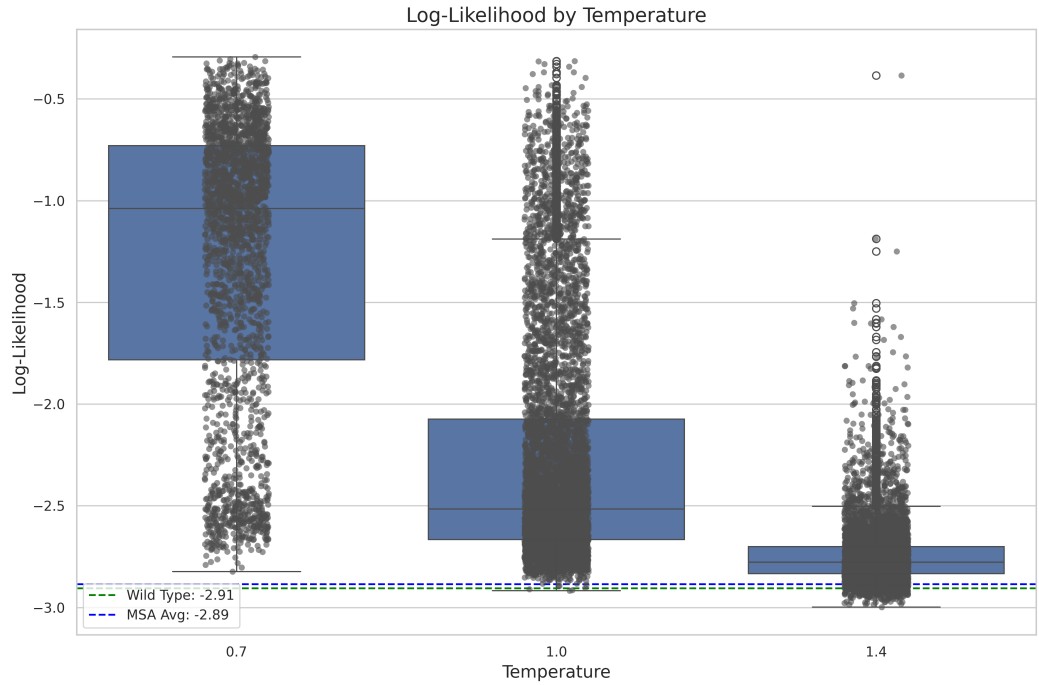

Figure 16: Log-likelihood versus selection of temperature for GFP.

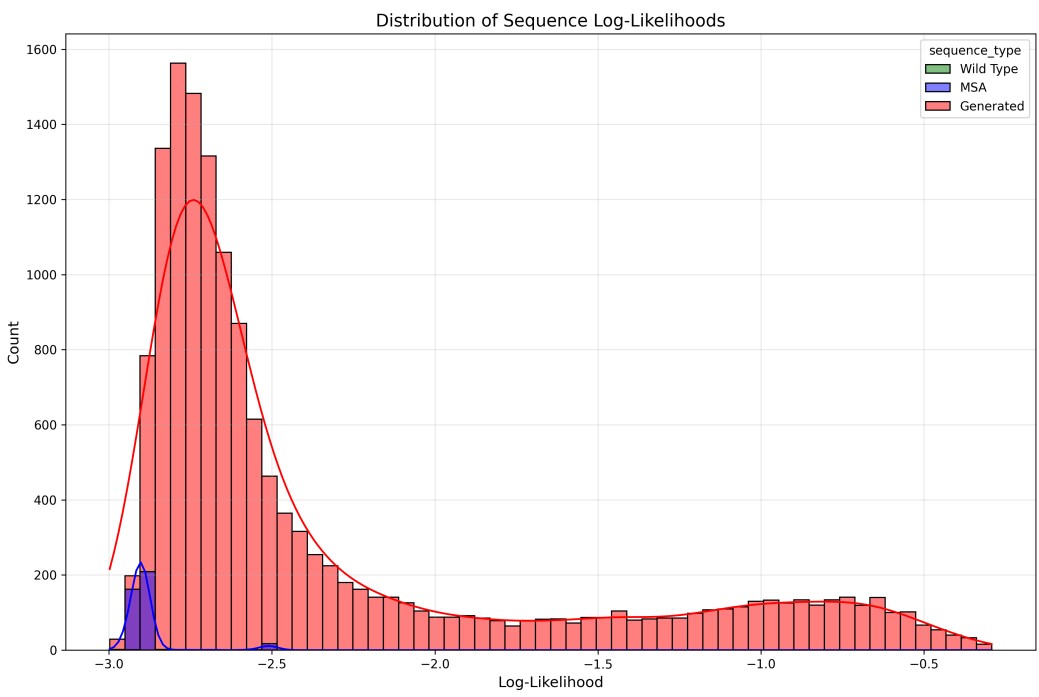

Figure 17: Likelihood distribution of GFP sequences generated using SpecMER compared to the likelihood distribution of its collected MSA (scored using ProGen2-M).

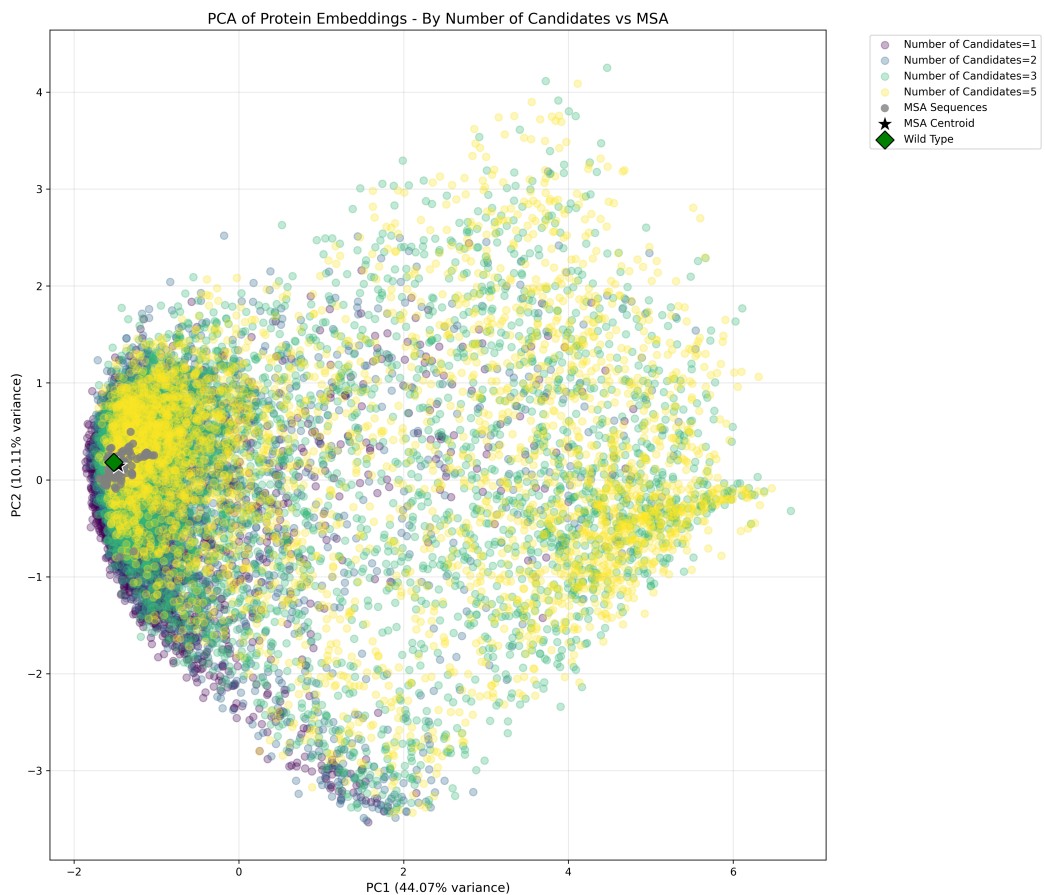

Figure 18: PCA plot for sequence embeddings generated with varying $c$, with $c = 1$ denoting speculative decoding, and $c > 1$ denoting SpecMER. Sequences are generated using the context of GFP.

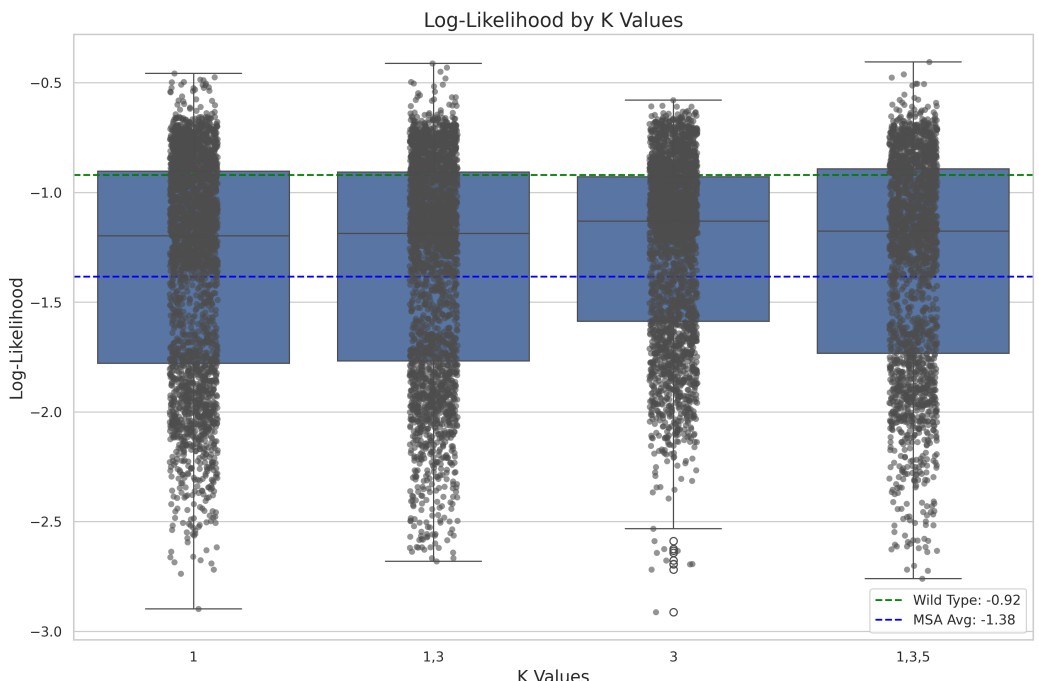

Figure 19: Log-likelihood versus selection of $k$ for Bgl3.

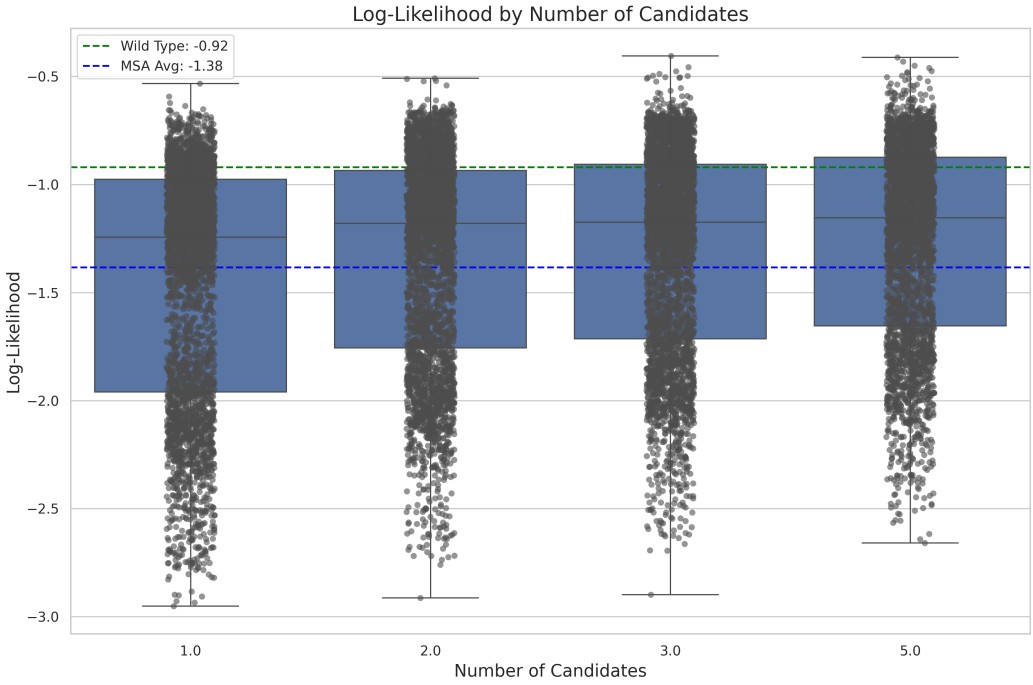

Figure 20: Log-likelihood versus selection of candidates $c$ for Bgl3.

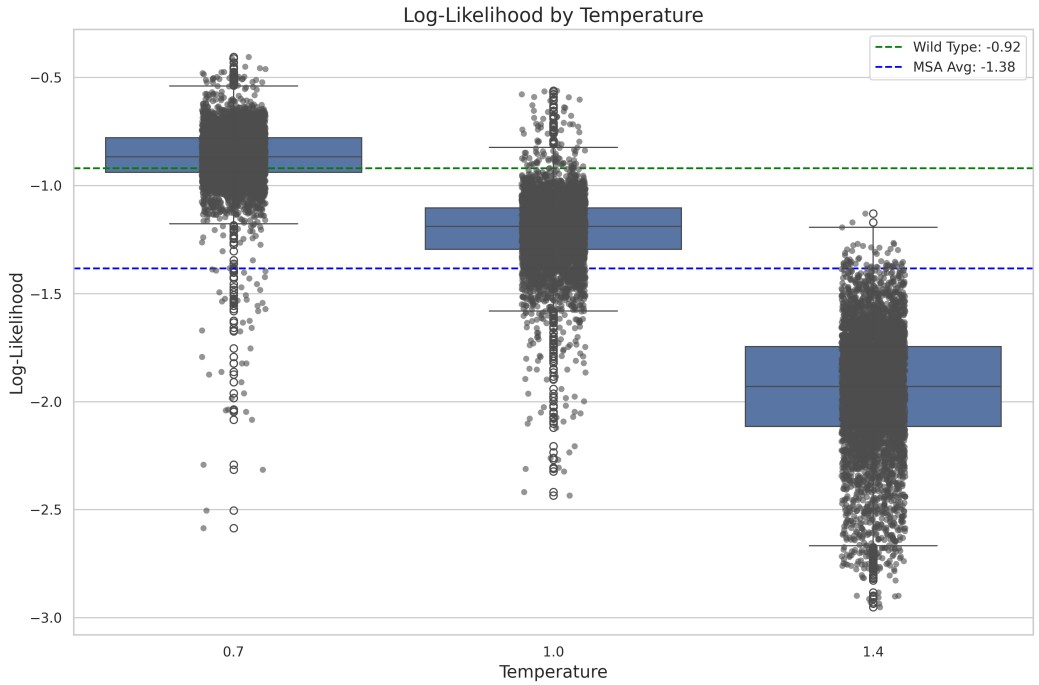

Figure 21: Log-likelihood versus selection of temperature for Bgl3.

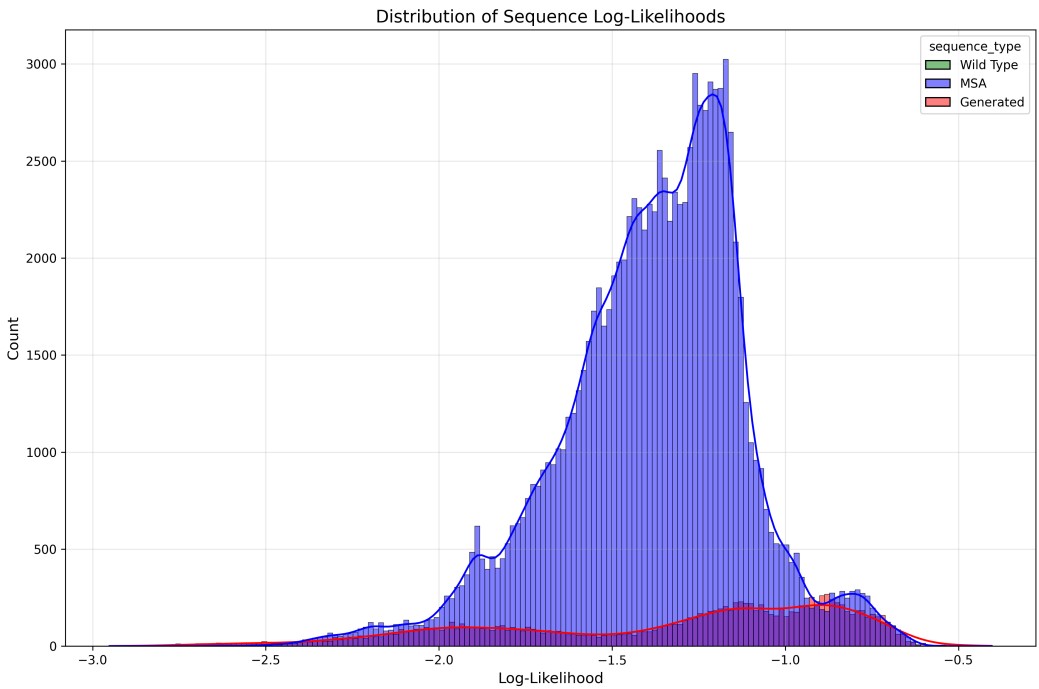

Figure 22: Likelihood distribution of Bgl3 sequences generated using SpecMER compared to the likelihood distribution of its collected MSA (scored using ProGen2-M).

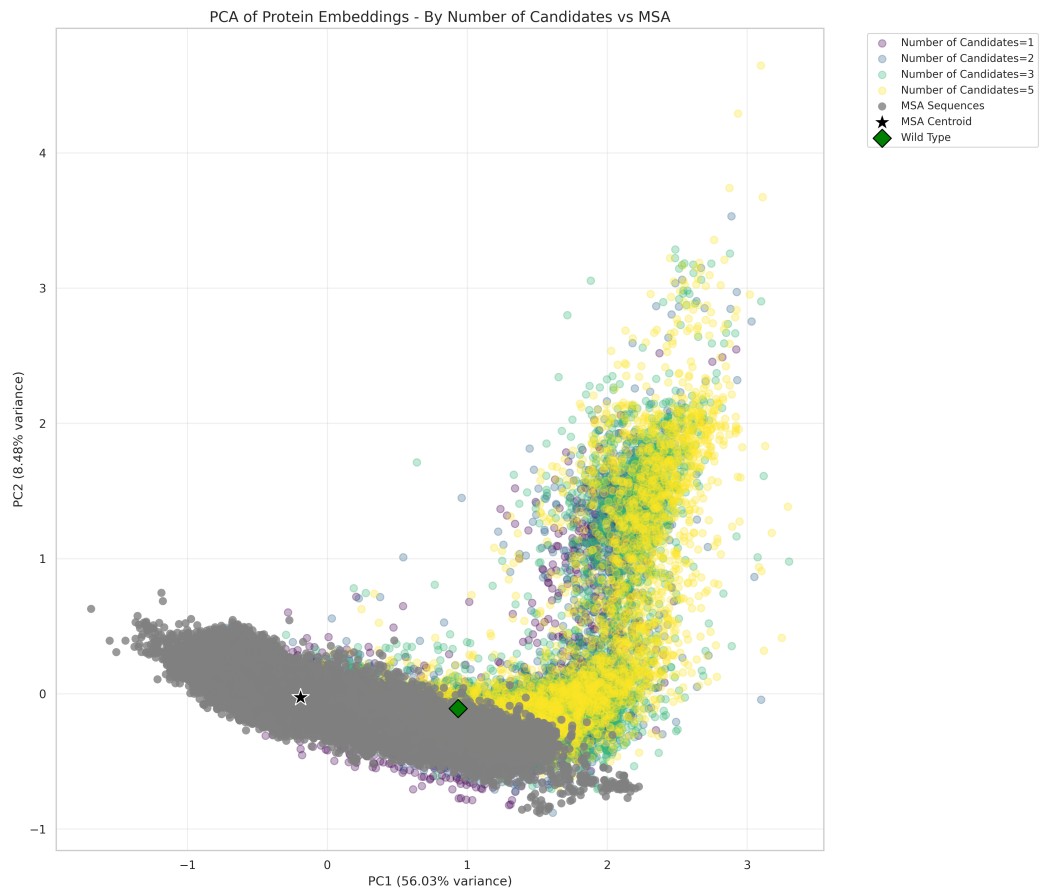

Figure 23: PCA plot for sequence embeddings generated with varying $c$, with $c = 1$ denoting speculative decoding, and $c > 1$ denoting SpecMER. Sequences are generated using the context of Bgl3.

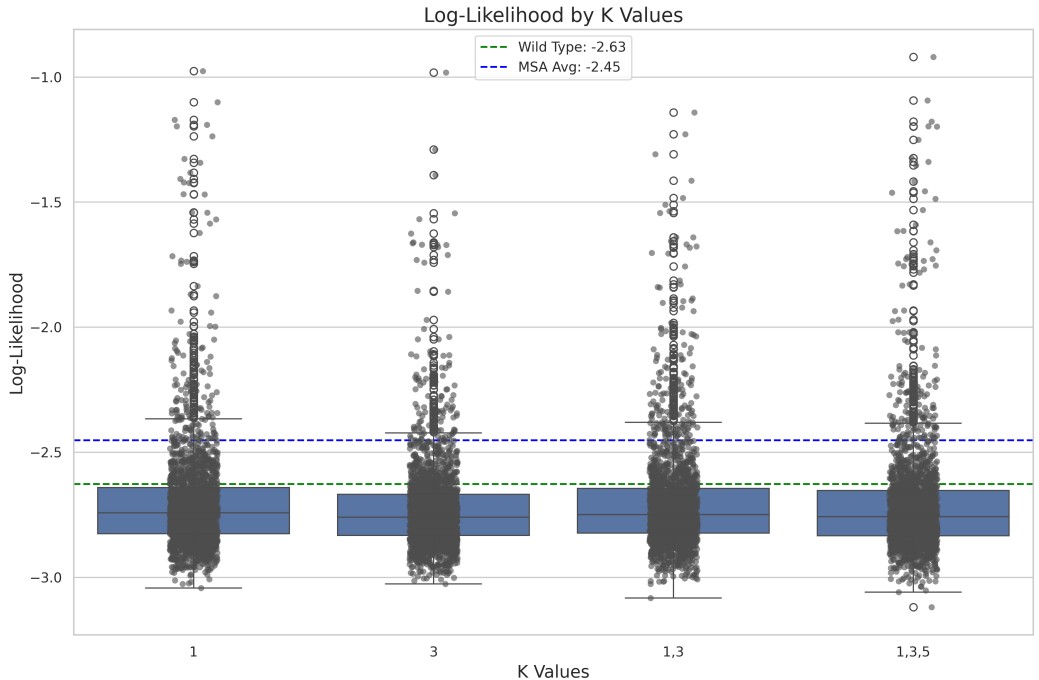

Figure 24: Log-likelihood versus selection of $k$ for RBP1.

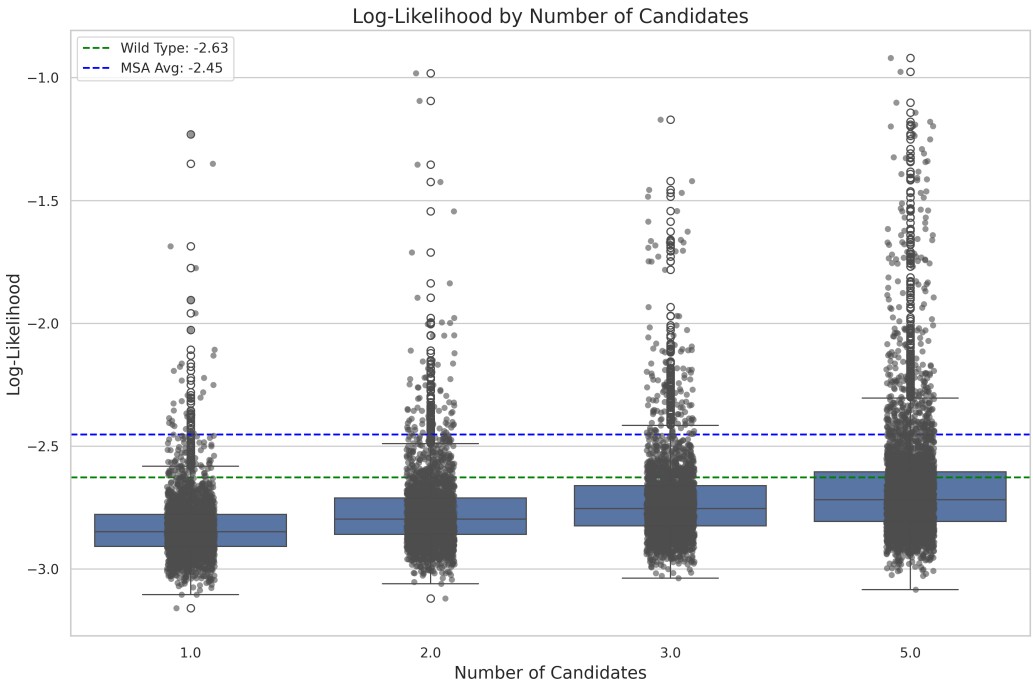

Figure 25: Log-likelihood versus selection of candidates $c$ for RBP1.

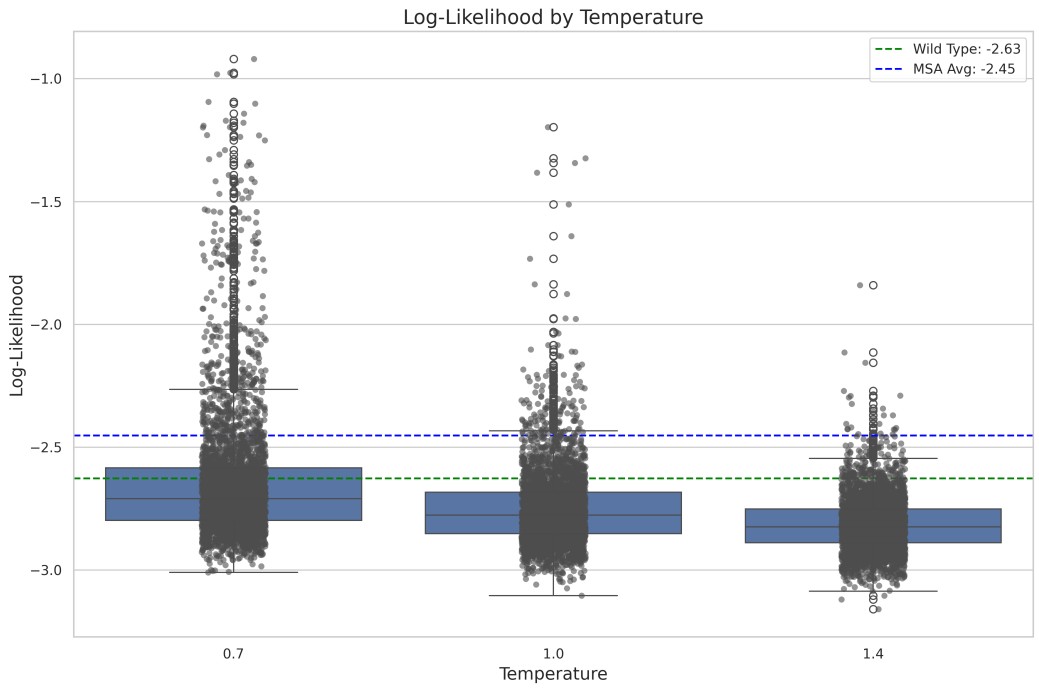

Figure 26: Log-likelihood versus selection of temperature for RBP1.

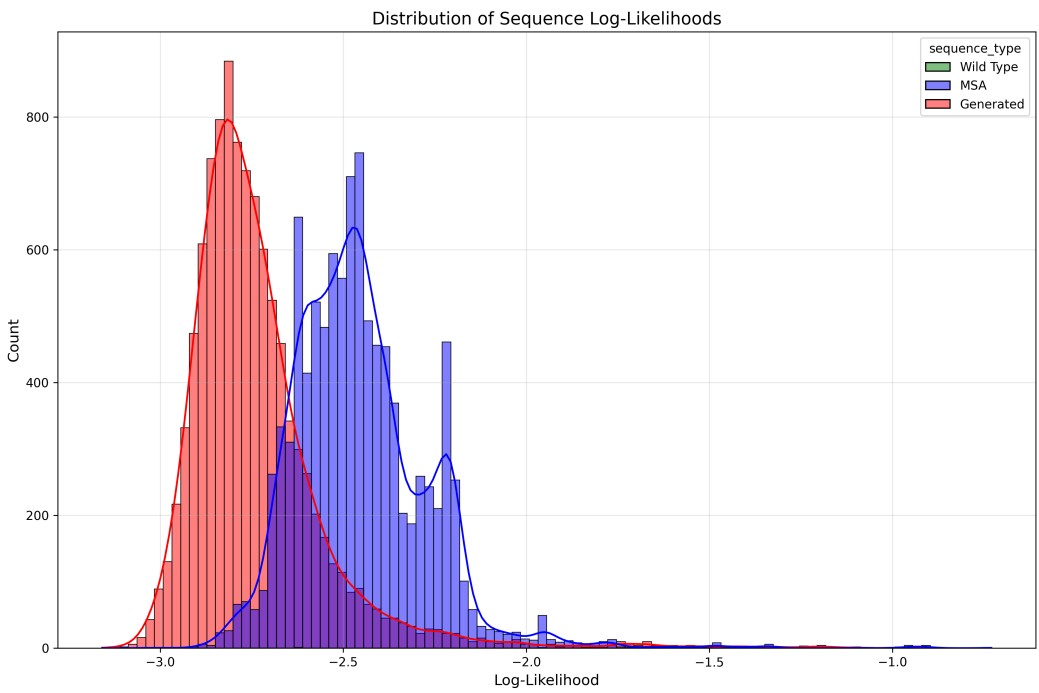

Figure 27: Likelihood distribution of RBP1 sequences generated using SpecMER compared to the likelihood distribution of its collected MSA (scored using ProGen2-M).

