# OpenReview forum: "SpecMER: Fast Protein Generation with K-mer Guided Speculative Decoding"
_NeurIPS.cc/2025/Conference — NeurIPS 2025 spotlight_

### Official Review · Reviewer_R8tH · 2025-06-09

**Clarity:** 3
**Significance:** 2
**Originality:** 3
**Rating:** 5
**Confidence:** 3

**Summary:**

The paper introduces SpecMER: an extension of speculative decoding from natural language processing to protein sequences. In addition to using a combination of draft and target models, SpecMER uses k-mer-based filtering to select candidate sequences generated by the draft model. The method is applied to five proteins from different functional categories and consistently shows a generation quality increase over the standard speculative decoding. Depending on the number of candidate sequences generated by the draft model, SpecMER obtains a speed up w.r.t. the target model of between 25-30%. The paper presents a thorough theoretical model analysis.

**Questions:**

1. L146-147: As per L140, the k-mer scoring function scores $c$ sequences of length $L$. In that case, shouldn’t the list be over the length $L$ sequences, i.e., $[\tilde{x}^{(1)}(t+L), \tilde{x}^{(2)}(t+L), …, \tilde{x}^{(c)}(t+L)]$? The shown list suggests to me that the function scores $c\times L$ sequences but perhaps I’m misunderstanding.
2. L178: Should be lowercase Q and P according to Algorithm 1.
3. Section 4: It is not immediately clear to me what the final hyperparameter configuration actually is. In section 4.2 and in the appendix, it is stated that all combinations of hyperparameters are evaluated. Does this mean that all 36 configurations are used to generate 200 sequences each, and the single configuration with the best results are then chosen and presented in the tables?
4. Regarding the hyperparameter sweep, I think it would benefit the reader to present the results quantitatively instead of in many different figures which can make it difficult to track the results.
5. SpecMER is intended to replace the standard autoregressive generation mechanism. While the speed-up is demonstrated in Table 4, I do not see any results comparing the *quality* of the generated sequences of SpecMER to the “vanilla” models. I think that experiments should be conducted and presented where the vanilla draft and target models are used generate a similar number of sequences (e.g, 36x200 sequences or in a similar amount of time) as SpecMER. Quality metrics like pLDDT and likelihoods could then be used to directly compare the model outputs. While a speed-up of ~30% is neat, it is not by itself convincing without proper comparison.
6. Is the termination criteria of stopping when the wild-type sequence length has been reached sensible? Or could this have negative effects on sequence (and the resulting structure) quality?
7. What would the metrics look like if the exact wild-type sequence was sampled?
8. Often when generating sequences conditioned on a reference (e.g., wild type), the aim is to generate diverse sequences with similar properties.  I think it could be interesting to include a diversity metric to see how far from the wild type and how far from each other the sampled sequences are. It could be as simple as computing the sequence identity between the reference and samples.
9. Three of the five proteins/domains examined are shorter than 100 amino acids. I think the inclusion of more sequence diversity would more clearly demonstrate the benefits of SpecMER.
10. What is the complexity of SpecMER  w.r.t. sequence length and hyperparameters?
11. How does MSA depth affect the quality of the generated sequences? Since only five proteins (three of which are very short) are examined, I’m not yet convinced that MSA depth and quality is independent.
12. Which ESM-2 version was used to compute the embeddings for the PCA? Were the embeddings mean-pooled? Are the aligned sequences embedded, or are they converted into unaligned sequences? I think some additional details should be included in appendix C.1.
13. How does context length affect generation quality? The context length is chosen to be approximately 10 % of the wild-type sequence length. Why is that?
14. In the introduction, ProGen2-XL is used as an example, yet the chosen target model is ProGen2-M. What would the speed up using SpecMER be if ProGen2-XL was used instead?

**Ethical Concerns:**

["NO or VERY MINOR ethics concerns only"]

**Final Justification:**

The authors have adequately addressed my raised questions, weaknesses, and limitations. I believe that the concept of speculative decoding is highly useful for protein sequence generation. I appreciate that the authors have gone beyond simply applying the method to protein data to adapting it to biological sequences with biological priors. I believe that the methods and results presented are of interest to the community and would therefore like to see the paper featured at this year's NeurIPS.

**Limitations:**

Provided that my raised weaknesses are addressed and questions answered, I think that the authors adequately address their work's limitations.

**Paper Formatting Concerns:**

No formatting issues.

**Quality:**

4

**Strengths And Weaknesses:**

Strengths

1. Biologically motivated extension of speculative decoding.
2. Thorough theoretical analyses.
3. Increased autoregressive generation speed.
4. Well-written paper.
5. Inclusion of code and data for reproducibility.

Weaknesses

1. Limited evaluation. The method is evaluated on five proteins, three of which are shorter than 100 amino acids. I think that a more diverse selection of proteins should be examined. I also think that the inclusion of a diversity metric (such as sequence identity or similarity to the wild type) or the inclusion of additional sequence quality metrics would further demonstrate the improved generation.
2. Limited comparison. The quality of the generated sequences are only compared between different hyperparameter configurations of SpecMER. Since the intention is to replace standard autoregressive decoding, I think that the generation quality should be directly compared to the sequences generated by the draft and target models, potentially at different temperature values.
3. Limited impact. Without the above analyses, the model presumably yields similar quality sequences ~30% faster. While this is certainly useful, the comprehensive hyperparameter sweep generates a total of 7200 sequences which is not insubstantial. I think that the benefits of using SpecMER compared to simply sampling from the target model at different temperatures for a comparable amount of time should be clearly demonstrated in order to convincingly show the impact.

I like the paper and would like to see it published, but the above weaknesses and related questions should be addressed and adequately and convincingly answered for me to raise my score.

---

> ### Author Rebuttal · Authors · 2025-07-31
>
> We thank you for your feedback. Your constructive comments have helped us clarify key points and strengthen our paper.
>
> > Limited evaluation. The method is evaluated on five proteins, three of which are shorter than 100 amino acids. I think that a more diverse selection of proteins should be examined. I also think that the inclusion of a diversity metric (such as sequence identity or similarity to the wild type) or the inclusion of additional sequence quality metrics would further demonstrate the improved generation.
>
> We have performed additional experiments on two longer proteins: CBS_HUMAN (551 amino acids) and ADRB2_HUMAN (413 amino acids). We report the average, top-20 and top-5 NLL:
>
> ADRB2_HUMAN
> Decoding Method|Average NLL|Top-20 NLL|Top-5 NLL
> -|-|-|-
> |Spec. Decoding| 1.90 +/- 0.65 | 1.18 +/- 0.30 | 0.78 +/- 0.16
> |SpecMER (c=3)| 1.33 +/- 0.50 | 0.77 +/- 0.13 | 0.61 +/- 0.04
> |SpecMER (c=5)| 1.03 +/- 0.60 | 0.57 +/- 0.11 | 0.43 +/- 0.04
>
> CBS_HUMAN
> Decoding Method|Average NLL|Top-20 NLL|Top-5 NLL
> -|-|-|-
> |Spec. Decoding| 2.42 +/- 0.42 | 2.06 +/- 0.44 | 1.43 +/- 0.38
> |SpecMER (c=3)| 2.17 +/- 0.66 | 1.47 +/- 0.50 | 0.84 +/- 0.16
> |SpecMER (c=5)| 1.87 +/- 0.68 | 1.14 +/- 0.32 | 0.74 +/- 0.07
>
> We found that SpecMER outperformed speculative decoding, further supporting our existing results. We will include these results in the main text.
>
> We agree that diversity metrics help strengthen our findings. For each protein, we computed the average Hamming distance between generated sequences and the wild-type sequence. Additionally, we computed the average pairwise Hamming distance between generated sequences to measure inter-sequence diversity:
>
> Protein | WT Distance (SpecMER) | WT Distance (Spec. Decoding) | Inter‑Seq (SpecMER) | Inter‑Seq (Spec. Decoding)
> -|-|-|-|-
>  | GFP | 208.35 ± 5.76 | 208.49 ± 4.91 | 181.78 | 184.56 ± 27.14 |
>  | RBP1 | 41.27 ± 3.48 | 42.81 ± 3.43 | 42.60 ± 3.87 | 44.88 ± 4.00 |
>  | ParD3 | 75.97 ± 3.01 | 78.68 ± 2.41 | 67.39 ± 6.87 | 70.00 ± 5.41 |
>  | GB1 | 44.70 ± 3.25 | 45.27 ± 3.27 | 46.47 ± 4.46 | 46.99 ± 3.86 |
>  | Bgl3 | 324.88 ± 30.88 | 333.12 ± 33.96 | 261.02 ± 42.28 | 284.97 ± 40.44 |
> | CBS | 378.64 +/- 140.60 | 431 +/- 106.05 | 291.84 +/- 161.60 | 457.27 +/- 40.98 |
> | ADRB2 | 263.80 +/- 120.22 | 290.18 +/- 109.01 | 270.82 +/- 93.46 | 340.28 +/- 47.66
>
> SpecMER generates sequences far away from the wild-type while maintaining inter-sequence diversity, exploring sequence space while still generating plausible protein sequences.
>
> > Limited comparison. The quality of the generated sequences are only compared between different hyperparameter configurations of SpecMER. Since the intention is to replace standard autoregressive decoding, I think that the generation quality should be directly compared to the sequences generated by the draft and target models, potentially at different temperature values.
>
> SpecMER, being a type of speculative decoding, increases the generation speed with no sacrifice to the quality of the sequences (Leviathan et al. (2021), Sun et al. (2022)). We discuss this property and expand on its result in section 2 and 3, but are happy to discuss in more detail in the camera ready version.
>
> To demonstrate this property, we conducted additional experiments generating sequences using only the target model (ProGen2-M) for 4 hours to compare sequence likelihoods to SpecMER (c=5) using the same temperatures as the results in Table 2. The resulting sequences were scored using ProGen2-M, and we report the top 20 highest likelihood averages:
>
> Protein|Target Model|SpecMER (c=5)
> -|-|-
> |Bgl3|0.78 +/- 0.02|0.63 +/- 0.11
> |GFP|0.51 +/- 0.04| 0.41 +/- 0.07
> |RBP1| 1.62 +/- 0.15| 1.72 +/- 0.30
> |GB1| 2.27 +/- 0.24| 2.20 +/- 0.31
> |ParD3| 0.69 +/- 0.11|0.67 +/- 0.12
>
> We found that protein sequences generated by SpecMER (c=5) had likelihoods comparable to, or exceeding, those from the target model, a result we discuss in Section 3.3.
>
> > Limited impact. Without the above analyses, the model presumably yields similar quality sequences ~30% faster. While this is certainly useful, the comprehensive hyperparameter sweep generates a total of 7200 sequences which is not insubstantial. I think that the benefits of using SpecMER compared to simply sampling from the target model at different temperatures for a comparable amount of time should be clearly demonstrated in order to convincingly show the impact.
>
> We perform an extensive hyperparameter sweep, which we detail in Appendix D, to identify the best configuration per protein, a one time cost that results in accelerated generation. As noted in our previous response, we performed additional experiments comparing SpecMER (c=5) to standard autoregressive decoding, using the same temperatures of the results in Table 2. SpecMER produced sequences with likelihoods comparable to or exceeding standard autoregressive generation, supporting that generation quality is maintained while achieving significant speedups.
>
> > L146-147: As per L140, the k-mer scoring function scores c sequences of length L. In that case, shouldn’t the list be over the length L  sequences, i.e., …,
>
> You are correct. The way it is described in the paper could be clarified. c x L indicates the batch dimension, i.e. c sequences each of length L.
>
> > Section 4: It is not immediately clear to me what the final hyperparameter configuration actually is. In section 4.2 and in the appendix, it is stated that all combinations of hyperparameters are evaluated.
>
> The final hyperparameter configurations for each protein reported in Table 2 are:
> Protein | Temperature | Draft Tokens | k value | Candidates
> -|-|-|-|-
> Bgl3| 1.0 | 5 | 3 | 5
> GFP| 0.7 | 5 | 1,3 | 5
> RBP1| 1.0 | 10 | 3 | 5
> GB1| 1.4 | 10 | 1,3,5 | 5
> ParD3| 1.0 | 5 | 1,3,5 | 5
> CBS| 0.7 | 5 | 1,3,5 | 5
> ADRB2| 0.7 | 5 | 1,3 | 5
>
> We will include these values in the Appendix.
>
> > Does this mean that all 36 configurations are used to generate 200 sequences each, and the single configuration with the best results are then chosen and presented in the tables?
>
> Yes.
>
> > Is the termination criteria of stopping when the wild-type sequence length has been reached sensible? Or could this have negative effects on sequence (and the resulting structure) quality?
>
> Yes. Our aim is to design variants around the wild-type sequence, a common practice in protein engineering (Hayes et al., 2025). In practice, we observed no degradation in sequence quality - sequences matched or exceeded the quality of those generated with standard autoregressive decoding.
>
> > What would the metrics look like if the exact wild-type sequence was sampled?
>
> Here is the negative log-likelihood and pLDDT for each wild-type sequence:
> Protein| NLL | pLDDT
> -|-|-
> CBS | 0.75 | -
> Bgl3 | 0.92 | -
> ADRB2 | 1.31 | -
> ParD3 | 2.11 | 0.79
> GB1 | 2.52 | 0.82
> RBP1 | 2.63 | 0.83
> GFP | 2.93 | 0.42
>
>  > What is the complexity of SpecMER w.r.t. sequence length and hyperparameters?
>
> SpecMER has the same computational complexity as speculative decoding, O(L^2), with respect to sequence length L. It does not introduce additional overhead beyond speculative decoding, whose accelerated generation is achieved without increasing asymptotic complexity relative to standard autoregressive decoding (Leviathan et al., 2021). The additional k-mer scoring in SpecMER is constant time and does not affect overall complexity. We will make this clear in the main text.
>
> > How does MSA depth affect the quality of the generated sequences? Since only five proteins (three of which are very short) are examined, I’m not yet convinced that MSA depth and quality is independent.
>
> We ran additional experiments to test the importance of MSA depth for Bgl3, forming k-mers using 1,000 sequences from the MSA instead of the full depth MSA (130k sequences), limiting the guidance signal from the MSA. We observed a sharp decline in likelihoods, with the top-20 sequences yielding a likelihood of 1.56 +/- 0.20, whereas SpecMER (c=5) with the full depth MSA yielded an average of 0.63 +/- 0.11. This supports our claim in the conclusion that performance may degrade when informative motifs are sparse or unavailable. We will include an updated section in the appendix with this ablation.
>
> > Which ESM-2 version was used to compute the embeddings for the PCA? Were the embeddings mean-pooled? Are the aligned sequences embedded, or are they converted into unaligned sequences? I think some additional details should be included in appendix C.1.
>
> We utilized ESM-2 8M with mean-pooled embeddings and unaligned sequences in accordance with standard practice. We will update Appendix C.1 to reflect this.
>
> > How does context length affect generation quality? The context length is chosen to be approximately 10 % of the wild-type sequence length. Why is that?
>
> We chose a context length of ~10% based on empirical observation: shorter contexts often led to pathological repetition of amino acids (Hsu et al., 2022), while longer contexts restricted exploration of sequence space. A 10% context provided a balance, enabling high-quality, diverse sequence generation.
>
> > In the introduction, ProGen2-XL is used as an example, yet the chosen target model is ProGen2-M. What would the speed up using SpecMER be if ProGen2-XL was used instead?
>
> We ran additional experiments with ProGen2-XL as the target model and ProGen2-M as the draft. The tokens per second jump from 7.03 to 9.72 with SpecMER (c=3), a 38% increase in speed.
>
> We will address your formatting concerns while making revisions. Thank you again for your feedback.

---

> > ### Comment · Reviewer_R8tH · 2025-08-05
> >
> > I appreciate the authors' rebuttals. My questions have been convincingly addressed and I will raise my score from 3 to 5. I hope that the additional results and analyses presented in the rebuttal will be part of the final manuscript.

---

### Official Review · Reviewer_icHu · 2025-06-16

**Clarity:** 4
**Significance:** 4
**Originality:** 3
**Rating:** 5
**Confidence:** 3

**Summary:**

In this paper, Speculative Decoding is introduced as a novel approach to sequence generation that accelerates standard autoregressive decoding techniques. Additionally, Speculative Decoding via k-mer Guidance is introduced to improve the efficiency of standard speculative decoding.

**Questions:**

Could you add more specific details relevant to the fact that this work is being submitted to a computer science conference? It currently feels like it lacks a clear description of the model architecture, training protocol, and computational resources used—I'd love to see that information, perhaps included in the appendix.

Also, looking at Table 4, it seems that increasing the c parameter in SpecMER leads to a noticeable increase in variance. Could you comment on why you think that might be the case? And what might be the concerns of using high c?

**Ethical Concerns:**

["NO or VERY MINOR ethics concerns only"]

**Final Justification:**

The authors have addressed my questions and provided the exact locations in the appendix where the needed information can be found

**Limitations:**

yes

**Paper Formatting Concerns:**

-

**Quality:**

3

**Strengths And Weaknesses:**

Strengths:

Strong empirical gains: 24–32% faster generation while producing sequences with higher likelihoods and better structural plausibility, validated across five diverse proteins.

Theoretical analysis of acceptance ratios and speedups is provided, and the open-source implementation makes the method accessible and reproducible.

Weaknesses:

It’s unclear what the main contribution of the paper is. From the abstract, it feels like SpecMER is the main contribution, which is built on top of existing speculative decoding methods. However, while reading the paper, if I understood correctly, speculative decoding was not used at all for protein generation. If this is the case, I think it should be emphasized more clearly in the abstract.

The following drawback follows from my previous comment: there are two metrics to evaluate model performance—speed and accuracy. However, there is a kind of inconsistency in the comparison. To show speed-up, you compare SpecMER with the target model and demonstrate improved performance, which is fair. However, in terms of accuracy, you compare only speculative decoding against SpecMER. If both of these methods are introduced by you, I don’t think this proves SpecMER’s efficiency convincingly. A comparison should also be made with existing methods—for instance, the one you used as a baseline for demonstrating the speed-up.

---

> ### Author Rebuttal · Authors · 2025-07-31
>
> Thank you for your constructive feedback. We have touched on some key points of clarification to address your comments.
>
> > It’s unclear what the main contribution of the paper is. From the abstract, it feels like SpecMER is the main contribution, which is built on top of existing speculative decoding methods. However, while reading the paper, if I understood correctly, speculative decoding was not used at all for protein generation. If this is the case, I think it should be emphasized more clearly in the abstract.
>
> To clarify our contributions:
>
> We adapt speculative decoding to protein language models and demonstrate acceleration over standard autoregressive decoding (Table 4, Baseline). Prior works in speculative decoding have focused on natural language. To the best of our knowledge, speculative decoding has not previously been applied to protein generation, so this is a key contribution of our work.
>
> We introduce SpecMER, maintaining speedups from speculative decoding, while generating protein sequences with higher biological plausibility.
> We discuss these contributions in lines 60-70, but we are happy to make it more clear in the abstract.
>
> > The following drawback follows from my previous comment: there are two metrics to evaluate model performance—speed and accuracy. However, there is a kind of inconsistency in the comparison. To show speed-up, you compare SpecMER with the target model and demonstrate improved performance, which is fair. However, in terms of accuracy, you compare only speculative decoding against SpecMER. If both of these methods are introduced by you, I don’t think this proves SpecMER’s efficiency convincingly. A comparison should also be made with existing methods—for instance, the one you used as a baseline for demonstrating the speed-up.
>
> SpecMER, being a type of speculative decoding, increases the generation speed with no sacrifice to the quality of the sequences (Leviathan et al. (2021), Sun et al. (2022)). We discuss this property and expand on its result in section 2 and 3, but are happy to discuss in more detail in the camera ready version.
> To demonstrate this property, we conducted additional experiments, generating sequences using only the target model (ProGen2-M) for 4 hours to compare sequence likelihoods to SpecMER (c=5). The resulting sequences were scored using ProGen2-M, and we report the top 20 highest likelihood averages:
> Protein|Target Model|SpecMER (c=5)
> -|-|-
> |Bgl3|0.78 +/- 0.02|0.63 +/- 0.11
> |GFP|0.51 +/- 0.04| 0.41 +/- 0.07
> |RBP1| 1.62 +/- 0.15| 1.72 +/- 0.30
> |GB1| 2.27 +/- 0.24| 2.20 +/- 0.31
> |ParD3| 0.69 +/- 0.11|0.67 +/- 0.12
>
> We found that protein sequences generated by SpecMER (c=5) had likelihoods comparable to, or exceeding, those from the target model, a result we discuss in Section 3.3.
>
> > Could you add more specific details relevant to the fact that this work is being submitted to a computer science conference? It currently feels like it lacks a clear description of the model architecture, training protocol, and computational resources used—I'd love to see that information, perhaps included in the appendix.
>
> ProGen2 is a decoder-only Transformer architecture with four different parameter sizes: small - 151M; medium/base - 764M; large - 2.7B; xlarge - 6.4B. No model training is performed in this work; inference is run on an NVIDIA A6000 GPU. More details can be found in Nijkamp et al. (2022). We discuss some of these details in Section 4, but are happy to add more details in the appendix.
>
> > Also, looking at Table 4, it seems that increasing the c parameter in SpecMER leads to a noticeable increase in variance. Could you comment on why you think that might be the case? And what might be the concerns of using high c?
>
> This increase in variance is due to batch generation with ProGen2. Batch generation is slightly slower than producing a single sequence continuation, and the added time to generate increases with c. Therefore, when a speculated token is rejected, decoding takes longer and the added cost of batch generation appears in the variance. Due to the non-deterministic nature of sampling, some generations have higher acceptance rates than others; therefore, occasional rises and dips in acceptance have a larger impact on tokens per second than producing a single sequence at a time. Further discussion on this trade-off space can be found in Appendix B.2.

---

> > ### Comment · Reviewer_icHu · 2025-08-06
> >
> > Thank you very much for your thoughtful responses and the clarifications you provided. After carefully considering your rebuttal, I am ready to re-evaluate my review and adjust my score to reflect a more positive assessment.

---

### Official Review · Reviewer_QzzJ · 2025-07-02

**Clarity:** 3
**Significance:** 2
**Originality:** 3
**Rating:** 4
**Confidence:** 3

**Summary:**

This paper presents SpecMER, a novel framework that accelerates autoregressive protein generation by enhancing speculative decoding. The method generates multiple draft sequences and uses k-mer frequencies from a Multiple Sequence Alignment (MSA) to select the most biologically plausible candidate for verification by a larger target model.

**Questions:**

Novelty is a key criterion in protein generation. Does incorporating MSA-derived information make the generated proteins more similar to existing sequences?

**Ethical Concerns:**

["NO or VERY MINOR ethics concerns only"]

**Final Justification:**

The authors have addressed my concerns.

**Limitations:**

yes

**Paper Formatting Concerns:**

No major formatting issues.

**Quality:**

3

**Strengths And Weaknesses:**

Strengths
- To the best of my knowledge, this is the first work to combine k-mer evolutionary priors with speculative decoding for protein language models.
- On five protein targets, the method reports 24–32 % wall-time speed-ups over plain autoregressive decoding.
- All hyperparameters are disclosed, and the authors provide open-source code, enabling full reproducibility.

Weaknesses

Major
- Because computational sequence generation already outpaces wet-lab synthesis and screening by several orders of magnitude, the claim that “slow sequence generation can delay high-throughput design workflows by days” appears overstated and insufficiently justified.
- Improvements in negative log-likelihood and pLDDT are modest, and their biological significance is not demonstrated.
- The study lacks ablations with random k-mer tables or shuffled MSAs (or MSA from another protein), making it hard to confirm that observed gains come from genuine evolutionary signal rather than extra sampling.

Minor
- Figure 8 title: embeddings - > embeddings
- Line 161:  in accordance to ->  in accordance with

---

> ### Author Rebuttal · Authors · 2025-07-31
>
> Thank you for your constructive feedback. After reviewing your comments, we have touched on some key points of clarification.
>
> > Because computational sequence generation already outpaces wet-lab synthesis and screening by several orders of magnitude, the claim that “slow sequence generation can delay high-throughput design workflows by days” appears overstated and insufficiently justified.
>
> The quality of designs chosen for wet-lab synthesis can drastically speed up the design process by proposing better variants, shortening or skipping design cycles altogether (Biswas et al., 2021). SpecMER addresses this by generating biologically plausible sequences with higher likelihoods and pLDDT scores than standard autoregressive decoding (Section 4.3). By producing more high-quality sequences in the same compute time, SpecMER increases the likelihood of finding stronger candidates earlier, ultimately reducing the experimental effort required during wet-lab synthesis.
>
> > Improvements in negative log-likelihood and pLDDT are modest, and their biological significance is not demonstrated.
>
> The goal of this work is to introduce a faster decoding method that can be applied to existing PLMs. We demonstrate that we can increase decoding speedups without degrading output quality, and in many instances, produce a better output. We believe this is a valuable contribution to the design process, particularly in scenarios where speed is critical (i.e. producing large libraries of candidate sequences).
> We note that even modest increases in negative log-likelihood and pLDDT scores will result in improved structural quality (Table 3) and can result in drastic improvement in function. Our results demonstrate that SpecMER achieves these improvements while enabling faster protein generation and accelerating protein design workflows.
>
> > The study lacks ablations with random k-mer tables or shuffled MSAs (or MSA from another protein), making it hard to confirm that observed gains come from genuine evolutionary signal rather than extra sampling.
>
> We performed two ablation studies to test the efficacy of k-mers. We first tested SpecMER using two configurations: generate conditioned on GFP, use GB1-derived k-mers to select continuations; generate conditioned on GB1, use Bgl3-derived k-mers to select continuations.
>
> Condition          | Mean NLL ± Std | Top‑20 Avg. NLL ± Std
> -|-|-
> | GFP + GB1 k‑mers   | −2.52 ± 0.27   | −1.78 ± 0.23         |
> | GB1 + Bgl3 k‑mers  | −2.79 ± 0.10   | −2.59 ± 0.11         |
>
> This mismatch in evolutionary signal led to lower likelihoods both on average and for top end generation (top-20) when compared to Table 2, indicating that protein specific k-mers are responsible for the increase in likelihood.
> The second ablation tested the importance of MSA depth. We generated Bgl3 proteins using 1,000 sequences from the MSA instead of the full depth MSA (130k sequences), limiting the number of k-mers to guide generation. We observed a sharp decline in likelihoods, with the top-20 sequences yielding a likelihood of 1.56 +/- 0.20, whereas SpecMER (c=5) with the full depth MSA yielded an average of 0.63 +/- 0.11. This further supports our claim in the conclusion that performance may degrade when informative motifs are sparse or unavailable. We will include an updated section in the appendix with this ablation.
>
> > Novelty is a key criterion in protein generation. Does incorporating MSA-derived information make the generated proteins more similar to existing sequences?
>
> To evaluate novelty, we computed the Hamming distance between each generated sequence and the wild-type, as well as the pairwise Hamming distance among generated sequences (see table below).
> Protein | WT Distance (SpecMER) | WT Distance (Spec. Decoding) | Inter‑Seq (SpecMER) | Inter‑Seq (Spec. Decoding)
> -|-|-|-|-
>  | GFP | 208.35 ± 5.76 | 208.49 ± 4.91 | 181.78 | 184.56 ± 27.14 |
>  | RBP1 | 41.27 ± 3.48 | 42.81 ± 3.43 | 42.60 ± 3.87 | 44.88 ± 4.00 |
>  | ParD3 | 75.97 ± 3.01 | 78.68 ± 2.41 | 67.39 ± 6.87 | 70.00 ± 5.41 |
>  | GB1 | 44.70 ± 3.25 | 45.27 ± 3.27 | 46.47 ± 4.46 | 46.99 ± 3.86 |
>  | Bgl3 | 324.88 ± 30.88 | 333.12 ± 33.96 | 261.02 ± 42.28 | 284.97 ± 40.44 |
>
> SpecMER generates sequences far from the wild-type while maintaining inter-sequence diversity, exploring novel sequence space. Additionally, Appendix D illustrates embedding plots comparing generated sequence to MSA sequences, indicating that SpecMER generates novel, high-likelihood sequences that are distinct from the wild-type yet remain grounded near homologous sequence space. We will update the appendix to include these results.

---

> > ### Comment · Reviewer_QzzJ · 2025-08-02
> >
> > I appreciate that the authors have addressed my concerns.

---

### Official Review · Reviewer_iBwT · 2025-07-08

**Clarity:** 3
**Significance:** 3
**Originality:** 3
**Rating:** 5
**Confidence:** 4

**Summary:**

Authors note that a limiting step in ML protein engineering is generation. Thus, they introduce SpecMER - a novel protein generation framework that accelerates autoregressive generation using speculative decoding guided by biologically-informed k-mer motifs from MSAs. By integrating these k-mer patterns, SpecMER prioritizes biologically plausible sequences during generation, improving both structural fidelity and functional plausibility without sacrificing speed. Across five diverse proteins, SpecMER achieves up to 32% faster generation compared to standard methods and consistently improves log-likelihoods and predicted structural confidence scores (pLDDT).

**Questions:**

1. How does ProGen-XL, -S, -M compare in speed to each other and how does that compare to other autoregressive models that exist? Having a 32% speedup is great, but if other AR models are already faster then wouldn't the protein designer just want to use those? No cross-model speed benchmarks are included, and no papers are cited that provide such benchmarks either.
2. Can you provide more experiments to back up the claim "The effectiveness of SpecMER depends on the quality of input alignment (MSA);
performance may degrade when informative motifs are sparse or unavailable"? It is intuitive, but not actually supported by your work. Right now, you have described the size of the MSAs for your different proteins of interest but there is not information on the "quality" of the MSA - are you talking about how well aligned everything is? the depth of the MSA? how many relevant proteins are included/excluded? I believe you just pulled the MSAs from ProteinGym, so perhaps consider assessing the quality and seeing if it leads to changes in performance in the model. Also consider testing multiple MSAs per protein (e.g. at different bitscores) and see what impacts performance.
3. There is a lot of talk about the tradeoffs of "cheap" generators generating a ton TON of sequences and then having a good filtering set that can quickly weed out the bad sequences. Have the authors considered this? Have you dont any comparison between this paradigm and the speculative decoding approach?

**Ethical Concerns:**

["NO or VERY MINOR ethics concerns only"]

**Final Justification:**

I recommend acceptance, and encourage the authors to expand on biological evaluation in future work or the camera-ready version.

**Limitations:**

yes

**Paper Formatting Concerns:**

Figure text is very small.

**Quality:**

3

**Strengths And Weaknesses:**

Quality:

Strengths

- The authors build a robust and thoughtful generative framework for protein sequence design that balances speed, plausibility, and computational cost.
- SpecMER is clearly motivated and well-engineered, combining speculative decoding with biological priors (k-mers) in a way that addresses their motivation
- The paper includes some empirical support: results across five functionally diverse proteins, ablation over multiple hyperparameters (e.g. number of candidates, k-mer size, temperature), and evaluation using both likelihood and structural metrics like pLDDT.

Weaknesses

- Authors should include a few additional citations to increase the quality of their scholarship and expand upon why generation from autoregressive models is challenging. Yes, it can take a while, but more than that there are some pathologies with that kind of generation that have been discussed in the literature:
(A) Cite Hsu et al. 2022, “Learning inverse folding from millions of predicted structures” (ESMF1) — they mention common autoregressive sampling failures, like pathological repetition (e.g., “EEEEEEE”).
(B) Additionally, Spinner et al. 2023, “How well do generative protein models generate?” outlines deeper issues with autoregressive models drifting from biologically meaningful sequences.
- The evaluation framework is lacking biological relevance. Have the authors checked if key residues in the proteins they design are preserved or accurately sampled? For example, the chromophore in GFP, catalytic residues in enzymes, or critical binding residues in GB1 and RBP1. Including this would make the biological quality claims more compelling.


Clarity:

Strengths
- Generally the writing in the paper is very clear. Authors do a good job of describing relevant hyper parameters

Weaknesses
- Figure text is very small and impossible read without magnification. Figure 2 is especially difficult to read without zooming in a lot.
- Figure 1B is important and difficult to compare the different plots - is there a way for the authors to overlay the traces on a single plot so that it's easier to compare by eye?


Significance:

Strengths
- This framework is very significant for the field of ML protein design.
- Authors highlight theoretical and architectural benefits of this speculative decoding. It's clear that this will impact many peoples work and potentially benefit generative protein design

Weaknesses
- But they do not provide concrete biological or design throughput outcomes...Authors could be a bit more clear about how important a 32% speedup is. If their argument is that 65 hours is a LONG time to generate 20K sequences ("Such slow sequence generation can delay high-throughput design workflows by days"), and a 32% speedup would mean that they could generate the same number of sequences in 40 hours (still days), does that matter/what does that actually get us? What is the tradeoff between 13K sequences at original speed vs 20K sequences at faster speeds? perhaps some simulation of the quality of the "best" top sequences from both experiments would be helpful to see that you are actually pushing the distrusting further to the right means


Originality:

Strengths
- The paper brings a clever and original twist to speculative decoding by incorporating biologically grounded priors (k-mers) from MSAs. People talk about generating more sequences and having better filtering as a way of getting around the time issue, and this approach more more creative

Weaknesses

- The speculative decoding setup itself is largely borrowed from prior work in NLP, and while applying it to protein generation is useful, it’s not a conceptual leap. Much of the novelty comes from the domain-specific k-mer filtering rather than a fundamentally new decoding strategy.

---

> ### Author Rebuttal · Authors · 2025-07-31
>
> Thank you for your thoughtful and detailed feedback. Your constructive comments have helped us clarify key points and strengthen the paper.
>
> > Authors should include a few additional citations to increase the quality of their scholarship and expand upon why generation from autoregressive models is challenging.
>
> We will include the following references:
> (Hsu et al., 2022),
> (Spinner et al., 2023),
> (Holtzman et al., 2020).
> Additionally, we will expand upon challenges in autoregressive decoding in the introduction, including pathological repetition, early termination, and OOD generation.
>
> > Have the authors checked if key residues in the proteins they design are preserved or accurately sampled?
>
> Our work does not explicitly evaluate the preservation of individual residues, as the core objective of SpecMER is to accelerate protein generation while maintaining or improving overall sequence quality. Our results demonstrate that proteins generated by SpecMER achieve higher likelihoods and pLDDT scores than those of speculative decoding, suggesting that the generated sequences align with commonly used design proxies.
>
> > What is the tradeoff between 13K sequences at original speed vs 20K sequences at faster speeds?
>
> SpecMER, being a type of speculative decoding, increases the generation speed with no sacrifice to the quality of the sequences; therefore, there is no inherent tradeoff (Leviathan et al. (2021), Sun et al. (2022)). We discuss this property and expand on its result in section 2 and 3, but are happy to discuss in more detail in the camera ready version.
> We conducted additional experiments generating sequences using only the target model (ProGen2-M) to compare sequence likelihoods to SpecMER (c=5). The resulting sequences were scored using ProGen2-M, and we report the top 20 highest likelihood averages:
> Protein|Target Model|SpecMER (c=5)
> -|-|-
> |Bgl3|0.78 +/- 0.02|0.63 +/- 0.11
> |GFP|0.51 +/- 0.04| 0.41 +/- 0.07
> |RBP1| 1.62 +/- 0.15| 1.72 +/- 0.30
> |GB1| 2.27 +/- 0.24| 2.20 +/- 0.31
> |ParD3| 0.69 +/- 0.11|0.67 +/- 0.12
> We found that sequences generated by SpecMER (c=5) had comparable or better likelihoods compared to standard autoregressive generation from the target model, a key property of speculative decoding discussed in Sections 2.1 and 3.3.
>
> > How does ProGen-XL, -S, -M compare in speed to each other and how does that compare to other autoregressive models that exist? Having a 32% speedup is great, but if other AR models are already faster then wouldn't the protein designer just want to use those? No cross-model speed benchmarks are included, and no papers are cited that provide such benchmarks either.
>
> We tested the generation speed of ProGen2-S, M, and XL. ProGen2-S: 74.11 toks/s, ProGen2-M: 31.48 toks/s, ProGen2-XL: 9.72 toks/s. We observed a sharp decline in speed, reinforcing the difficulties of generating large amounts of sequence with autoregressive generation. We then performed experiments with Tranception and found that it generated sequences more slowly than ProGen2 and with lower likelihoods. As SpecMER is a decoding strategy for accelerating autoregressive generation, it is applicable to any AR model given a compatible vocabulary size.
>
> > Can you provide more experiments to back up the claim "The effectiveness of SpecMER depends on the quality of input alignment (MSA); performance may degrade when informative motifs are sparse or unavailable"?
>
> We performed two ablation studies to test the efficacy of k-mers. We first tested SpecMER using two configurations: generate GFP, use GB1-derived k-mers to select continuations; generate GB1, use Bgl3-derived k-mers to select continuations.
> Condition          | Mean NLL ± Std | Top‑20 Avg. NLL ± Std
> -|-|-
> | GFP + GB1 k‑mers   | −2.52 ± 0.27   | −1.78 ± 0.23         |
> | GB1 + Bgl3 k‑mers  | −2.79 ± 0.10   | −2.59 ± 0.11         |
>
> This mismatch in evolutionary signal led to lower likelihoods both on average and for top end generation (top-20) when compared to Table 2, indicating that protein specific k-mers are responsible for the increase in likelihood.
> The second ablation tested the importance of MSA depth. We generated Bgl3 proteins using 1,000 sequences from the MSA instead of the full depth MSA (130k sequences), limiting the number of k-mers to guide generation. We observed a sharp decline in likelihoods, with the top-20 sequences having a likelihood of 1.56 +/- 0.20, whereas SpecMER (c=5) with the full depth MSA had an average of 0.63 +/- 0.11. This further supports our claim in the conclusion that performance may degrade when informative motifs are sparse or unavailable. We will include an updated section in the appendix with this ablation.
>
> > There is a lot of talk about the tradeoffs of "cheap" generators generating a TON of sequences and then having a good filtering set that can quickly weed out the bad sequences. Have the authors considered this?
>
> We have not considered this filtering strategy. The goal of SpecMER is to design many high quality sequences, using k-mers as a form of filtering during generation, and we view our method as adjacent to this line of work.
> We will address all of your formatting concerns while making revisions. Thank you again for your feedback.

---

> ### Author Response · Authors · 2025-08-08
>
> Thank you again for your constructive feedback. Please let us know if there is any remaining point you would like us to clarify or expand upon before the discussion period ends.

---

### Note · Authors · 2025-08-12

Following the conclusion of the rebuttal period, we would like to highlight key points of discussion:

> We received many comments inquiring about the target model's performance compared to SpecMER.

We addressed these comments with a discussion on key properties of speculative decoding, which ensure generated sequences are consistent with outputs from the target model. Additionally, we ran experiments to demonstrate this property, illustrating that in some cases, SpecMER outperforms target-only decoding for top-end sequence metrics.

> Reviews asked about the importance of MSA depth and quality.

We addressed these comments with two ablations, quantifying the impact that k-mers have on generation quality. We tested the impact of generating sequences with SpecMER using k-mers from a different protein, finding that sequence likelihoods steeply declined. Additionally, we reduced the size of a large MSA (Bgl3), finding that the resulting generated sequences exhibited lower likelihoods on average.

> The inclusion of a sequence diversity metric was suggested.

We computed the Hamming distance between generated sequences and their respective wild-type sequence, highlighting the novelty of sequences generated with SpecMER. Additionally, we computed the Hamming distance between generated sequences to demonstrate inter-sequence diversity. Following comments on a lack of longer proteins, we included two more proteins with sequence lengths > 400 to strengthen our findings.

We appreciate the thoroughness of the reviews. We have addressed each of the reviewers' concerns to the best of our ability, and will include the results of these discussions in the appropriate sections of the camera-ready paper.

---

### Decision · Program_Chairs · 2025-09-17

**Decision:**

Accept (spotlight)

**Comment:**

This submission introduces SpecMER, a speculative decoding framework for accelerating autoregressive protein generation, integrating biological priors (k-mer motifs extracted from MSAs) into the decoding process, guiding a lightweight draft model to propose biologically plausible sequences that a larger target model then verifies. This balances speed (via speculative decoding) and sequence quality (via MSA-derived k-mer filtering), addressing the latency bottleneck of standard autoregressive models in high-throughput protein design.

The strengths noted by the reviewers include its first domain adaptation of speculative decoding to protein generation, strong empirical performance (24–32% speedups across five initial proteins, extended to two longer proteins >400 amino acids during rebuttal), and extensive validation metrics. The method’s biological grounding by using MSA k-mers to avoid implausible sequences was also emphasized as a key advantage over generic speculative decoding. During the rebuttal period, the authors also addressed reviewer concerns thoroughly. Although limitations such as the need for more functional validation (e.g., key residue preservation) and broader testing on diverse protein families remain, all the reviewers are in favor of this submission and I also agree. We strongly encourage the authors to incorporate the additional results and valuable discussions in the revised manuscript.